# Mapping effective connectivity of human amygdala subdivisions with intracranial stimulation

Masahiro Sawada[1,2], Ralph Adolphs [3], Brian J. Dlouhy [1,4], Rick L. Jenison [5], Ariane E. Rhone[1], Christopher K. Kovach [1], Jeremy, D. W. Greenlee [1,4], Matthew A. Howard III[1,4,6] & Hiroyuki Oya [1,4] ✉

The primate amygdala is a complex consisting of over a dozen nuclei that have been implicated in a host of cognitive functions, individual differences, and psychiatric illnesses. These functions are implemented through distinct connectivity profiles, which have been documented in animals but remain largely unknown in humans. Here we present results from 25 neurosurgical patients who had concurrent electrical stimulation of the amygdala with intracranial electroencephalography (electrical stimulation tract-tracing; es-TT), or fMRI (electrical stimulation fMRI; es-fMRI), methods providing strong inferences about effective connectivity of amygdala subdivisions with the rest of the brain. We quantified functional connectivity with medial and lateral amygdala, the temporal order of these connections on the timescale of milliseconds, and also detail second-order effective connectivity among the key nodes. These findings provide a uniquely detailed characterization of human amygdala functional connectivity that will inform functional neuroimaging studies in healthy and clinical populations.

The amygdala is implicated in a host of emotional and cognitive functions and dysfunctions[1] and is a crucial component for establishing conditioned and unconditioned behavioral responses in both humans[2] and animals[3]. A number of specific functions have been linked to the amygdala, including decision-making, face perception, declarative memory, fear, and control of breathing. These behaviors have been associated with connectivity between the amygdala and prefrontal cortex[4], temporal cortex[5], hippocampus[6], prefrontal cortex[7], and brainstem nuclei[8–10], respectively. As with functions in healthy individuals, psychiatric dysfunction depends on the complex connectivity between the amygdala and other brain structures[11–13]. This information is critically important, yet difficult to obtain in humans.

Studies in rodents and non-human primates have shown that the mammalian amygdala is comprised of multiple subnuclei that differ in their function and patterns of projections via a large network with other brain regions[14,15], implementing high-dimensional representations that can be used to flexibly guide behavior[16]. The parcellation and connection of the amygdala were first studied by using anatomical neural tracers in rats and cats[17–22] and in non-human primates (NHP)[23–25]. Based on these anatomical studies, it is generally accepted that the lateral nucleus of the amygdala is the major site receiving inputs from sensory cortices (although the medial nucleus of the amygdala notably receives direct olfactory input as well), whereas the central nucleus is an output region whose projections include brainstem. Thus, the subnuclei-level connectivity profiles are essential to understand the functional role of the amygdala[3,26]. Of note, the primate amygdala is in fact comprised of over a dozen nuclei, beyond the resolution of MRI in humans. In the present study, we acknowledge the

[1]Department of Neurosurgery, Carver College of Medicine, University of Iowa, Iowa City, IA, USA. [2]Department of Neurosurgery, Tazuke Kofukai Medical Research Institute and Kitano Hospital, Osaka, Japan. [3]Division of Humanities and Social Sciences, California Institute of Technology, Pasadena, CA, USA. [4]Iowa Neuroscience Institute, University of Iowa, Iowa City, IA, USA. [5]Department of Neuroscience, University of Wisconsin - Madison, Madison, WI, USA. [6]Pappajohn Biomedical Institute, University of Iowa, Iowa City, IA, USA. ✉e-mail: hiroyuki-oya@uiowa.edu

limitations in resolution by referring to groups of nuclei: we refer to the medial group and the lateral group throughout. The lateral group comprises the lateral nucleus of the amygdala (the largest in the human amygdala), and the medial group all parts of the amygdala medial to that (see Fig. 1d and "Methods").

Structural connectivity studies show comparable (but not identical) patterns of connectivity as does functional connectivity, data for the latter in humans is typically inferred from functional MRI[27].

Analyses of tracer studies in cats situate the amygdala as a hub in a very broad network of regions with which it is structurally connected[28]. Similarly, resting-state functional connectivity from BOLD-fMRI reveals correlated activity with much of neocortex[29], which can be selectively disrupted with pharmacological (NHP)[30] or electrical (humans)[31] perturbation of the amygdala. While there are now MRI-based structural atlases of the human amygdala that provide considerably improved spatial resolution of the major nuclei, at least at the

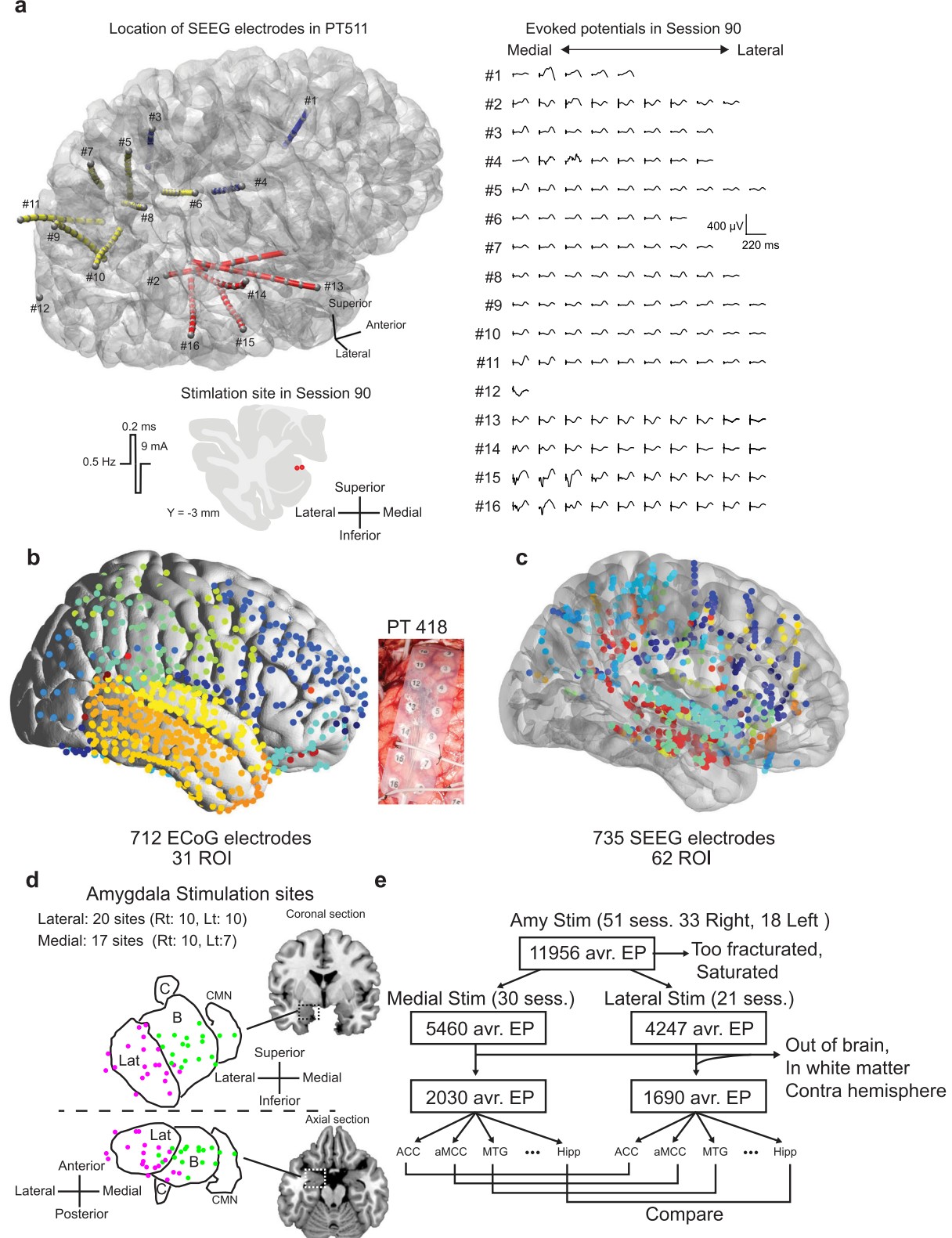

**Fig. 1 | Electrical stimulation tract tracing (es-TT) in epilepsy patients.**
**a** Representative evoked potentials (EPs) in a patient (PT 511), SEEG (stereotactic electroencephalography) electrodes were inserted from frontal (bule), parietal (yellow) and temporal (red) lobe. Red dots indicate the stimulation site in the amygdala. Stimulus waveform is also shown. **b** Distribution of subdural ECoG (electrocorticography) electrodes plotted on MNI template brain. All electrodes were classified according Destrieux ROI that are color-coded. Middle picture shows subdural ECoG grid taken during an implantation surgery. **c** Distribution of SEEG electrodes. Same as (**b**). **d** Projection drawing of the amygdala in the coronal and axial planes. Lat; lateral nucleus, B; Basal nucleus complex includes dorsal and intermediate subnuclei of basolateral nucleus, and basomedial nucleus. C; central nucleus, CMN; cortical and medial nucleus. Magenta dots represent the pair of

contacts' midpoints which were used for the lateral group amygdala stimulation. Green dots indicate the midpoint location for medial group amygdala stimulation. The boundary line was placed on the medial edge of lateral nucleus. **e** Flowchart showing the selection process of evoked potentials for the comparison analysis. Amy; amygdala, Stim; stimulation, avr. EP; averaged evoked potential, sess; session. For the creation of brain backgrounds in panels (**a**)–(**d**), we used template ICBM152 Nonlinear brain obtained from http://nist.mni.mcgill.ca/icbm-152-nonlinear-atlases-2009/under Copyright (C) 1993–2004 Louis Collins, McConnell Brain Imaging Centre, Montreal Neurological Institute, McGill University and Conte69.32k surface mesh[103] obtained from https://biomedia.doc.ic.ac.uk/brain-parcellationsurvey/) under WU-Minn HCP Consortium Open Access Data Use Terms.

group level[32–34], distinguishing their functional connectivity profiles has been more challenging. In addition to using diffusion-weighted MRI[35,36], resting-state functional connectivity has been used to parcellate the amygdala's nuclei and define the networks in which they participate[37–39], but this approach suffers again from relatively low resolution and difficulty in distinguishing functional connectivity (i.e., correlations in activity that could be mediated by complex network-level effects) from effective connectivity (direct causal influence)[40].

More unambiguous delineation of cause-effect relationships in the brain can be performed using perturbational techniques[41]. Focal brief electrical stimulation delivered with precise timing in combination with concurrent electrophysiological recording from distal brain sites can overcome the limitations inherent in other methods to produce detailed maps of effective connectivity. Electrical stimulation–tract tracing (es-TT) exactly employs this concept and offers fine timing information at the millisecond scale. Information obtained from such studies can indicate not only direct connections between the stimulated site and the response sites, but also contains information about how multiple responsive sites are communicating with each other, that is, the 'second-order connectivity that defines complex functional networks. With respect to the amygdala, alterations in such network-level function are thought to be a hallmark of psychiatric illness, including autism[42,43], anxiety disorders[12,44], depression[44,45], and schizophrenia[46], yet direct evidence in human subjects is lacking.

Aims of the current study were to provide an initial foundation, from two convergent approaches (es-TT and electrical stimulation with concurrent fMRI, es-fMRI), sampling broad regions of the brain. Our focus was to identify the effective connectivity of medial and lateral subdivisions (group of nuclei) of the human amygdala, from the spatial and temporal distribution of responses evoked by bipolar intracranial electrical stimulation. We first present electrophysiological recordings, supplemented by blood oxygenation-level-dependent (BOLD) functional MRI measurements. We combined both techniques to investigate connectivity of amygdala subdivisions in 25 patients with epilepsy. We not only examined first-order effects (putatively direct effective connectivity) but also second-order network-level connectivity in the brain.

We identified characteristic electrophysiological responses to electrical stimulation of human amygdala subdivisions with latencies that were centered at 15 and 150 ms, which provided the basis for subsequent analyses. To detect subtle but stable differential information contained in the broadly distributed response, we utilized non-linear machine learning models. Differential response patterns were reliably detected in cortical regions recorded (anterior cingulate cortex, precuneus, lateral prefrontal cortex, sensorimotor cortex, and superior and middle temporal lobe). The anatomical distribution of response was similar with es-fMRI. We also identified the temporal propagation of responses across multiple regions: the effects began in anterior cingulate cortex and orbitofrontal cortex, and propagated to prefrontal cortex before finally reaching sensorimotor cortex and parietal cortex. Second-order connectivity analyses modeled the directional connectivity among all these as well as additional brain

structures to provide a description of network-level functional connectivity. This detailed and relatively comprehensive characterization can provide the substrate for the interpretation of future neuroimaging studies examining individual differences and alterations in psychiatric illness.

## Results

### Electrical stimulation–tract tracing (es-TT) of medial and lateral groups of the amygdala

We stimulated 37 amygdala sites through depth electrodes in 13 epilepsy patients (Supplementary Table 1) while concurrently recording iEEG (intracranial electroencephalography) from a combination of contacts on other depth electrodes, as well as subdural strip and grid contacts placed over cortex. Representative electrode placement and electrical stimulation-averaged evoked potentials (EPs) in one of the subjects is shown in Fig. 1a. Comprehensive electrode locations were mapped onto a template brain and are shown in Fig. 1b, c. iEEG coverage included amygdala (depth electrodes), hippocampus (depth electrodes), cingulate gyrus (depth electrodes), orbitofrontal cortex (depth and subdural strips electrodes), wide regions of lateral frontal, temporal and parietal cortex including pre- and post-central gyrus, superior and middle temporal gyrus (subdural strips and grid electrodes). Stimulation sites were distributed within the lateral (La), basolateral (BL), basomedial (or accessory basal, BM), and cortical and medial (CMN) nuclei of the amygdala as shown in Fig. 1d. Note that CMN in the CIT168 atlas corresponds to the superficial nuclei group. We divided all these amygdala subnuclei into medial and lateral group according to an atlas of in vivo human amygdala parcellation (see "Methods"). Overall, we evaluated 11956 EPs and retained 3720 EPs after rejecting EPs originating from the seizure onset zone or recorded in the white matter. We analyzed 2030 and 1690 EPs for medial and lateral group stimulation, respectively, over 51 es-TT runs (30 and 21 es-TT runs for medial and lateral group, respectively) (Fig. 1d, e, Supplementary Table 1).

Observed EPs in response to electrical stimulation of the amygdala showed two prominent deflections in early (10–30 ms) and late (70–200 ms) time windows (region of interest (ROI) averaged waveform shown in Fig. 2a and Supplementary Fig. 1, ROI are from Destrieux's parcellation[47], see "Methods"). The peak of the late component had positive polarity and was located approximately at 150 ms after stimulation onset (hence referred to as P150). The peak of the early component varied in polarity and occurred ~15 ms after stimulus onset (N/P15). We evaluated the magnitude of responses in these two temporal windows by calculating the range-values, the difference between maximum and minimum amplitude (Fig. 2b, d, insets, also see "Methods") within each of these windows. Supra-threshold amplitude distributions for the early N/P15 (Fig. 2b) and late P150 (Fig. 2d) EPs were widespread, including anterior and posterior cingulate cortex (ACC and PCC), superior and middle temporal gyrus (STG and MTG), pre- and post-central gyrus (PreCG and PostCG), inferior, middle and superior frontal gyrus (IFG, MFG, and SFG), fusiform gyrus (FG), supramarginal gyrus (SMG), angular gyrus (AG), superior parietal

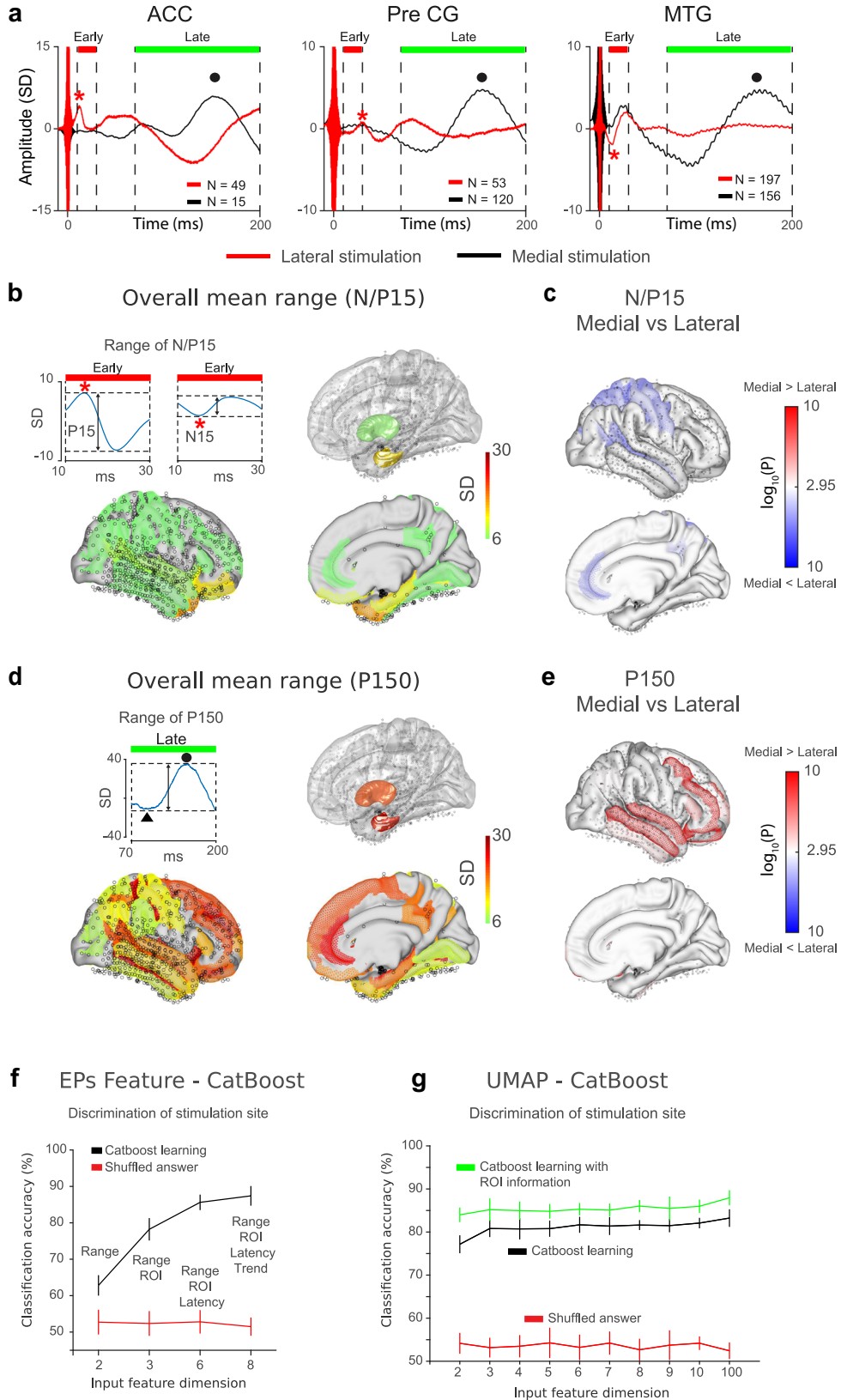

lobule (SPL), hippocampus, insula and orbitofrontal cortex (OFC), (Table 1).

There was a general tendency for medial group stimulation to elicit greater P150 amplitude whereas lateral amygdala stimulation showed larger N/P15 responses, as seen in Fig. 2a, for representative ROIs (See Supplementary Fig. 1 and Supplementary Table 2 for other

anatomical ROIs). To further elucidate the difference in EPs waveforms between the two stimulation locations, we applied two machine learning classification analyses to determine whether the patterns of EPs across all the es-TT runs can predict amygdala stimulation site (medial or lateral group stimulation). For the first classification analysis, input features we used were from quantification of each electrode's

**Fig. 2 | Spatial distribution of intracranial evoked potentials (EPs) to the electrical stimulation of the amygdala. a** Representative EPs waveforms averaged within the ROIs from all patients normalized with respect to baseline standard deviations plotted separately for medial- and lateral- group amygdala stimulation. Here, only mean waveforms are shown. Supplementary Fig. 1 shows mean and SE for all ROIs. Red asterisks and black dots on the waveform represent early (N/P15) and late (P150) components of EPs. Horizontal bar in red and green represent time windows for EPs component detection. **b** Spatial distribution of significant early EPs (N/P15) represented on the colormap on the MNI template brain. Insets indicate the method for detecting the range values. Color represents normalized EPs amplitude measured as range value in SD unit. Red asterisks indicate N/P15 peak. **c** Spatial distribution of anatomical ROIs that showed significant differential amplitude between medial and lateral amygdala stimulation ($P < 0.05$, two-sided $t$-tests, Bonferroni correction). Color represents $-\log_{10}(P\text{-values})$. **d** Same as panel (**b**), but for later EPs component (P150). Black triangle and dot in the inset indicate peak and valley

in the later time window defining P150 potential. **e** Same as panel (**c**), but for P150 component. Two-sided $t$-tests. **f** Decoding accuracy for stimulation site (medial- vs lateral-group amygdala stimulation). Horizontal axis indicates the input feature dimension. Here the input features were from EPs quantification process (see "Methods"). Range means the EPs range values (see panels **b** and **d** and "Methods"). Error bars indicate mean (center of error bars) and SE from 10-fold cross validation. Black and red line indicates classification accuracy from original and shuffled datasets, respectively. **g** Decoding accuracy of medial- and lateral-group amygdala stimulation performed on the EPs waveform. Time dimension was nonlinearly reduced with UMAP. Horizontal axis indicates the UMAP dimension used. Error bars and line-color indicate the same as in (**f**). Green line indicates classification accuracy of the original dataset but with ROI is also used as an input features. For the creation of brain backgrounds in panels (**b**)–(**e**), we used Conte69.32k surface mesh[103] obtained from https://biomedia.doc.ic.ac.uk/brain-parcellationsurvey/) under WU-Minn HCP Consortium Open Access Data Use Terms.

## Table 1 | Abbreviation table

| Brain ROIs | |
|---|---|
| STS | Superior temporal sulcus |
| STG | Suprior temporal gyrus |
| MTG | Middle temporal gyrus |
| ITG | Inferior temporal gyrus |
| ITS | Inferior temporal sulcus |
| OFC | Orbitofrontal cortex |
| PreCG | Precentral gyrus |
| PostCG | Postcentral gyrus |
| ACC | Anterior cingulate cortex |
| MCC | Middle cingulate cortex |
| aMCC | Anterior part of middle cingulate cortex |
| PCC | Posterior cingulate cortex |
| dPCC | Dorsal part of posterior cingulate cortex |
| SFG | Superior frontal gyrus |
| MFG | Middle frontal gyrus |
| IFG | Inferior frontal gyrus |
| SMG | Supramarginal gyrus |
| AG | Angular gyrus |
| SPL | Superior parietal lobule |
| FG | Fusiform gyrus |
| **ROI grouping** | |
| OF | Orbitofrontal group (Orbital gyri) |
| lPFC | Lateral prefrontal cortex group (SFG, MFG, IFG) |
| PL | Parietal group (AG, SMG, SPL, postCG) |
| TL | Temporal group (STG, MTG, ITG, STS, ITS) |
| CC | Cingulate group (ACC, dPCC) |
| SM | Sensorimotor group (PreCG, postCG, central sulcus) |
| **Methodology** | |
| es-TT | Electrical stimulation–tract tracing |
| es-fMRI | Electrical stimulation–functional MRI |
| ECOG | Electrocorticography |
| SEEG | Stereotactic electroencephalography |
| iEEG | Intracranial electroencephalography |
| EPs | Evoked potentials |
| N/P15 | Early component of EPs |
| P150 | Late component of EPs |
| CGC | Conditional Granger causality |
| UMAP | Uniform Manifold Approximation and Projection |
| FD | Frame-wise displacement |

EP response (see "Methods"), that is, the range values, the ROI identity, the peak latency for both N/P15 and P150, and the trend values (voltage difference between the first and last time point within the time windows). We sequentially added the features to the classifier and calculated the prediction accuracies shown in Fig. 2f. The analysis of the data from 3720 EPs using the Catboost algorithm[48] showed over 80% prediction accuracy with 5 features (range values and latencies within early and late time window, and ROI identity, 10-fold cross validation, Fig. 2f) indicating the existence of robust response profiles distributed over the brain sites we sampled that could serve to differentiate the amygdala regions stimulated.

We also performed a second classification analysis on continuous waveform of each contact's EPs, but with a non-linear dimensionality reduction, UMAP[49]. We treated each EPs time axis as a dimension that was reduced with UMAP. Following the dimensionality reduction, we subsequently carried out Catboost classifications with incremental number of UMAP features (from 2 to 100 UMAP dimensions, see "Methods"). The result again showed over 80% prediction accuracy for stimulation site classification (10-fold cross validation, Fig. 2g) using only these 2-dimensional UMAP features. As shown in Fig. 2g, inclusion of (categorical) ROI information in the model improved the classification accuracy. We then statistically evaluated where in the brain differences in response magnitude (Fig. 2c, e) differentiated the stimulation site (medial versus lateral group) for each ROI. Significant differences ($t$-test, with Bonferroni correction, $P < 0.05$) were observed in ACC, IPS (intraparietal sulcus), STS (superior temporal sulcus), MFG, SPL, preCG, postCG, precentral sulcus and OFC in the P/N15, and in STG, MTG, inferior temporal sulcus, SFG, MFG, IFG, and OFC in the P150 component (Fig. 2a–e, also see Supplementally Table 2 for their magnitude values).

### Electrical stimulation-fMRI (es-fMRI) of medial and lateral group of amygdala

We conducted concurrent electrical stimulation and functional MRI (es-fMRI) experiments on 16 epilepsy patients (4 overlapped with es-TT; see "Methods" and Supplementary Table 1) to examine differential response patterns following lateral versus medial group stimulation from BOLD responses measured in both hemispheres. The es-fMRI was conducted without having subjects engaged in any active task using a block design (es-ON and es-OFF) with charge-balanced square pulse electrical stimulation at 9–12 mA current as described previously[31]. Out of 30 valid amygdala stimulation es-fMRI runs, 20 runs were from medial, and 10 runs were from lateral group subdivisions (total of 25 stimulation locations in the amygdala, Fig. 3a). First-level GLM analyses showed robust positive as well as negative BOLD responses in individual runs (Fig. 3b). Second-level group analysis again showed statistically significant activations as well as deactivations (cluster

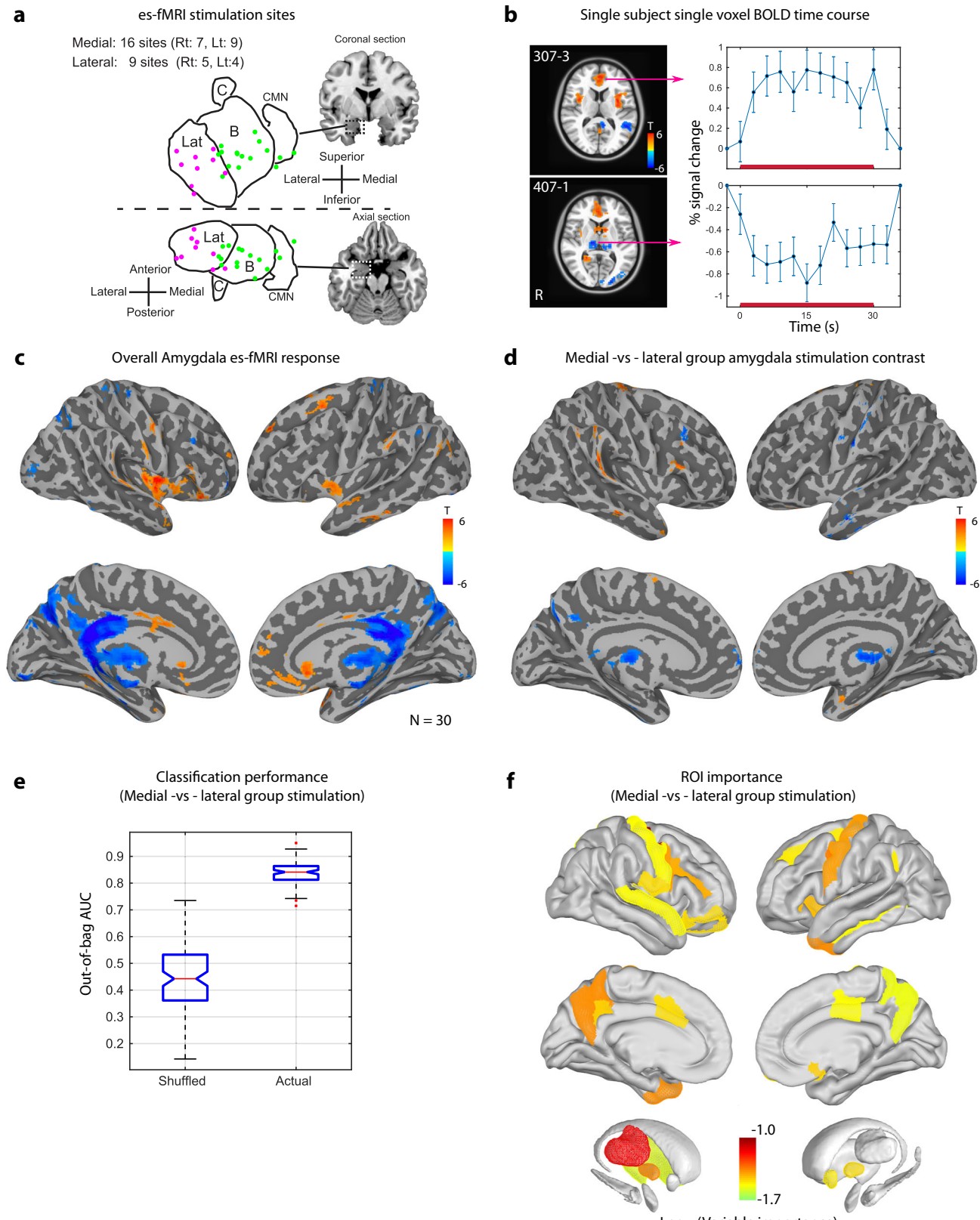

**a** es-fMRI stimulation sites

Medial: 16 sites (Rt: 7, Lt: 9)
Lateral: 9 sites (Rt: 5, Lt:4)

**b** Single subject single voxel BOLD time course

**c** Overall Amygdala es-fMRI response

N = 30

**d** Medial -vs - lateral group amygdala stimulation contrast

**e** Classification performance
(Medial -vs - lateral group stimulation)

**f** ROI importance
(Medial -vs - lateral group stimulation)

Log₁₀ (Variable importance)

alpha = 0.02 with primary voxel-wise threshold $P = 0.001$, Fig. 3c and Supplementary Fig. 2). Brain areas that showed significant BOLD responses included ACC, anterior middle cingulate cortex (aMCC), PCC, thalamus, precuneus, insula, FG, SFG, preCG, postCG, AG, and OFC (Supplementary Table 3). These regions overlapped considerably with those found in the es-TT experiments described above.

The second-level contrast analysis examining the differential response between medial group and lateral group stimulation showed that a number of focal anatomical sites exhibited significant difference (Fig. 3d) including in Thalamus, preCG, postCG, SFG, MFG, IFG, Precuneus, AG, STG, ITG, and inferior parietal lobule (Supplementary Table 3). Again, brain sites that showed differential responses of

**Fig. 3 | Amygdala stimulation es-fMRI responses. a** Stimulation sites for es-fMRI shown in colored dots mapped onto right amygdala. Blue and red dots represent midpoint of two adjacent amygdala stimulation contacts for medial (green dots) and lateral group (magenta dots), respectively. Lat: lateral nucleus, B: Basal nucleus complex + basomedial nucleus. C: Central nucleus and CMN: cortical and medial nucleus. Number of stimulation sites (Rt. Right amygdala and Lt. left amygdala) are shown separately for medial group and lateral group stimulation. **b** Representative single voxel BOLD response time course in two patients. Thresholded at $T = 3$ and cluster size of 70. Stimulation period is 0 to 30 s and is shown with red horizontal bars above x-axis. Error bars were from 10 es-ON blocks in the es-fMRI run. Mean and SE are shown. **c** Results from group analysis (multi-level mixed-effect analysis). $N = 30$ es-fMRI runs over 16 patients, 25 sites stimulated). Resulted statistical map was thresholded (cluster-wise multiple comparison correction through simulation; 3dClustSim, two-sided) at cluster size alpha = 0.02 with primary voxel-wise threshold at $P = 0.01$. **d** Group contrast results, medial versus lateral amygdala stimulation. The same threshold was used. See Supplementary Fig. 2 for statistical maps presented in the template volume.

**e** Multivariate classifier result for medial–vs–lateral group stimulation classification. Out-of-bag (oob) area under the receiver operating characteristic curve for original and shuffled data are shown. Edges of box indicate 25 and 75 percentile points along with medians indicated by red horizontal lines. Whiskers indicate minima and maxima. Outliers were indicated by red dots. Width of the notch was defined by $3.14 \times (75 \text{ percentile points} - 25 \text{ percentile point})/\sqrt{N}$. $N = 100$ (Number of classification run) for both actual and shuffled datasets. **f** ROI distribution that showed significant variable importance (above 95 percentile point of variable importance obtained from shuffled datasets) obtained from the classification above is shown. Color represents $\log_{10}$(variable importance). For the creation of brain backgrounds in panels (**a**)–(**d**) and (**f**), we used template ICBM152 Nonlinear brain obtained from http://nist.mni.mcgill.ca/icbm-152-nonlinear-atlases-2009/ under Copyright (C) 1993–2004 Louis Collins, McConnell Brain Imaging Centre, Montreal Neurological Institute, McGill University and Conte69.32k surface mesh[103] obtained from https://biomedia.doc.ic.ac.uk/brain-parcellationsurvey/) under WU-Minn HCP Consortium Open Access Data Use Terms.

medial group vs lateral group stimulation in es-fMRI overlapped with the results from es-TT experiments, even though these experiments were done on largely different patients. We also applied classification analysis using mean BOLD response within each anatomical ROIs (total 167 ROIs, from the Destrieux Atlas, and subcortical segmentation from FreeSurfer) across individual runs to examine whether stimulation condition in each run (medial group vs lateral group) can be predicted. Bagging on ensemble classification trees achieved significant classification performance (out-of-bag area under the receiver operating characteristic curve, oob-AUC > 0.83, Fig. 3e). Out-of-bag variable importance values from the classifier showed an anatomical distribution similar to what we had found in the es-TT experiment (Fig. 3f), including preCG, precentral sulcus, aMCC (anterior medial cingulate cortex), ITG and STG, as well as other regions such as thalamus, pallidum, precuneus, and insula.

## EP latency analysis: early N/P 15 component

The classification analyses showed that peak latency information is important for classification of stimulation site (medial vs lateral group stimulation, Fig. 2f). Definition of the latency for N/P 15 and P150 is shown in Fig. 4a and Fig. 5a. Latency distribution in one es-TT run shows short N/P15 latency in the orbitofrontal and anterior cingulate cortex (Fig. 4b).

For the population analysis, we grouped recording locations into these 6 groups: Orbitofrontal group (OF: orbital gyri, straight gyrus, and olfactory sulcus and orbital sulci), Lateral prefrontal cortex group (lPFC: SFG, MFG, inferior frontal sulcus, middle frontal sulcus, inferior and superior part of precentral sulcus), Sensorimotor group (SM: PreCG, PostCG, and central sulcus), Parietal group (PL: AG, SMG, SPL, intraparietal sulcus and post-central sulcus), Temporal group (TL: STG, MTG, ITG, STS, inferior temporal sulcus), and Cingulate group (CC: ACC and dorsal PCC; dPCC).

The peak latency of the N/P15 component in the lateral amygdala stimulation group showed significant differences across brain sites (Fig. 4c, one-way ANOVA, $P < 10^{-6}$). The OF and the CC group showed the shortest peak latency around 16–17 ms (OF: $17.0 \pm 0.5$ ms, CC: $16.6 \pm 0.8$ ms, mean and SE) followed by the temporal and frontal sites (SFG, MFG and STG, MTG) with the longest latencies in the parietal lobe ($20.8 \pm 0.6$ ms) (Fig. 4c). Post hoc tests showed that the contrast of OF–SM (Tukey's test, $P = 0.011$), OF–PL ($P = 4.69 \times 10^{-6}$), lPFC–PL ($P = 0.003$), PL–TL ($P = 6.12 \times 10^{-4}$), and PL–CC ($P = 4.71 \times 10^{-4}$) were significant (Fig. 4c). While ACC had shorter mean peak latency than PCC, given the limited number of contacts in PCC, this site difference in mean latencies did not reach statistical significance (two-sided $t$-test, $P = 0.17$, Supplementary Fig. 4a). This suggests that the propagation pattern of the initial response (N/P 15) progress from OFC and ACC to prefrontal and lateral temporal cortex, before reaching parietal cortex.

Findings based on this latency analysis suggest there is potential grouping in the brain based on the timing of response after amygdala stimulation. We tested this hypothesis with UMAP dimensionality reduction followed by the Catboost classification, which found boundaries that discriminated the three categories with high prediction accuracy using 2 UMAP dimensions. (Fig. 4d, 593 EPs were used, OF, lPFC, CC group = 82%, Parietal and sensorimotor (PL, SM group) = 52% and Temporal group = 80% accuracy). Intra-group EPs confirmed this latency relationship (Fig. 4e). The fast latency group (OF, CC, and lPFC group) showed an initial mean latency of 14.4 ms followed by TL group (16.9 ms) and PL and SM group (17.6 ms). Medial group stimulation also showed significant latency differences among cortical targets (Supplementary Fig. 3a, one-way ANOVA, $P < 10^{-10}$). Again, there was a tendency for the OF, CC, and lPFC groups to have shorter peak latencies than other cortical groups (see Supplementary Fig. 3a for statistical assessments). Note that N/P15 amplitudes were generally smaller with medial amygdala group stimulation than with lateral amygdala group stimulation (Fig. 2c).

## EP latency analysis: late P150 component

Next, we analyzed peak latency of the P150 component resulting from medial amygdala group stimulation (Fig. 5a). In a representative patient shown in Fig. 5b, ACC contacts showed earliest P150 peak latency, which were followed by responses in frontal lobe (STG, MTG, and PreCG) and then temporal and parietal lobe (STG, MTG, AG, and Precuneus) (Fig. 5b). In another participant with a high-density subdural surface grid over the temporal lobe, P150 latency showed spatial patterns that followed underlying anatomy; inferior temporal gyrus (ITG) and frontal cortex (preCG) showed the shortest latencies followed by the STG, and MTG showed the longest P150 latency within the sampled temporal lobe (Fig. 5c).

Sampling over all patients, the P150 peak latencies differed significantly across ROIs (one-way ANOVA, $P < 10^{-10}$). The population data demonstrated that the average peak latency in the Temporal group was the longest (Fig. 5d, mean $157 \pm 1.0$ ms, mean and SEM), whereas that in the Orbitofrontal group was the shortest ($129 \pm 6.8$ ms) followed by the intermediate Cingulate group ($143.3 \pm 2.6$ ms). We found significantly shorter latency in the ACC than in the dPCC (Supplementary Fig. 4b, ACC: $140.1 \pm 3.3$ ms, dPCC: $151.8 \pm 2.5$ ms, $P = 0.04$). Similar to the results from N/P15 responses, Post hoc analyses showed OF–lPFC (Tukey's test, $P = 1.71 \times 10^{-4}$), OF–SM ($P = 2.17 \times 10^{-8}$), OF–PL ($P = 2.66 \times 10^{-7}$), OF–TL ($P = 2.07 \times 10^{-8}$), OF–CC ($P = 0.006$) contrasts were all significantly different in their peak latency distributions. Also, lPFC–TL ($P = 2.83 \times 10^{-7}$), PL–TL ($P = 1.31 \times 10^{-4}$), and TL–CC ($P = 3.61 \times 10^{-6}$) contrasts were significant (Fig. 5d). These results again suggest there is a potential grouping in the P150 component based on its latency: a short-latency group (OF, Prefrontal cortex and Cingulate group), a middle latency

group (Sensorimotor and parietal group) and a long latency group (Temporal lobe). The UMAP followed by the Catboost classification again found boundaries that discriminated the three categories with high prediction accuracy using 2 UMAP dimensions (678 EPs were used, OF, lPFC, CC group = 57%, Parietal and sensorimotor (PL, SM group) = 87% and Temporal group = 80% accuracy). Intra-group averaged EPs confirmed the latency relationship above: P150 peak latency was shortest in OF, lPFC, CC groups (147 ms), followed by PL + SM group (157 ms), and longest in the temporal group groups (167 ms, Fig. 5f). The pattern of latencies in lateral amygdala stimulation was again such that OF and CC groups showed shorter latencies than the others (Supplementary Fig. 3b). Post hoc Tukey's tests showed significant differences in OF−PL ($P = 0.006$), lPFC−PL ($P = 0.013$), and PL−TL ($P = 0.019$) contrasts.

### Effect of hemispheric laterality and sex

Next, we examined whether the response differed significantly due to two important factors, sex and hemisphere[50–52]. Although these contrasts were pre-planned, given the limited number of subjects available (male patients $N = 6$, female patients $N = 7$, left hemisphere session $N = 7$, and right hemisphere session $N = 13$), here we report only the effect size calculated within each ROI groups of interest, rather than making statistical inferences. The p150 response was generally larger in male patients than female patients but there was no clear tendency for a hemispheric difference (Supplementary Fig. 8). While underpowered given the small subject sample, this analysis suggests that potential presence of sex-hemisphere interactions that will be important to follow up in future studies. The results show larger amygdala connectivity in the right hemisphere in men, and conversely, women showed larger connectivity in the left hemisphere, especially in the early time window (N/P15). This effect was most pronounced in OF, CC, TL, and SM groups. Note that our data concerns connectivity from the amygdala and not the responses within the amygdala. Thus, these preliminary results provide initial data to extend findings on amygdala laterality from local activation to brain-wide connectivity.

### Response correlations to patients' psychiatric condition: depression and anxiety

We also explored possible associations between patients' psychiatric condition and amygdala connectivity found by es-TT. We identified the patients who had diagnosis of either depression or anxiety, and compared the es-TT response between presence ($N = 5$) and absence ($N = 8$) of such conditions (see Supplementary Table 1). Using all stimulation combined (medial + lateral amygdala group), SFG, MFG, IFG, STG, MTG, insula, postcentral sulcus, and AG showed significant differences between the two psychiatrically defined conditions (Supplementary Fig. 9a, c, $P < 0.05$, Bonferroni correction). Interestingly, the frontal cortex showed a greater response and temporal lobe showed smaller response in the positive-symptom group. If all testable ROIs were combined, no significant difference was detected (Supplementary Fig. 9b and 9c). These exploratory analyses were not pre-planned and will need to be followed up in future studies with larger samples.

### Second-order connectivity from local field potentials

The above distributions of EP latency and magnitude are a first-order effect of brain perturbation. We further examined information flow between the different sites using non-parametric Conditional Granger Causality analysis (CGC, see "Methods") in three patients who had multiple es-TT runs and electrode coverage in most of the ROI groups of interest (PT384, 511, and 515). We hypothesized that second-order correlations would be most prominent in later time windows and thus used data from 70 to 200 ms post amygdala stimulation. Figure 6 shows CGC between those pairs of ROIs that had significant connectivity induced by medial group stimulation. The effective connectivity was significant and bidirectional between the cingulate cortex and temporal lobe group and lPFC group (CC ↔ lPFC and CC ↔ TL group). Weaker but significant connectivity was also found in PL → lPFC group, SM → lPFC group, and CC → SM group (Fig. 6a–c). Another CGC analysis on a different patient confirmed stronger connectivity within the anterior part of cortex (among CC, OF, and lPFC group) and weaker but significant CGC to more posterior parts of cortex (TL and PL group). Note that the recording location among these patients varied due to clinically directed electrode placement. The CGC analysis revealed that there was a peak of CGC within the alpha frequency range (around 8 Hz), which presumably reflects the P150 component in the EPs (Fig. 6a). This analysis indicated that the flow of stimulation-induced effective connectivity generates a network with ACC and lateral prefrontal cortices as the key nodes.

Compared to medial amygdala group stimulation, CGC for lateral amygdala group stimulation (shown in Supplementary Fig. 7) showed a marked decrease in ACC−lPFC connectivity (becoming non-significant), while ACC−OFC connectivity remained equivalently strong. TL-CC connectivity also showed reduced connectivity.

### Second-order connectivity from BOLD-fMRI

We also investigated second-order connectivity using the es-fMRI data with BOLD residual time series correlations (see "Methods"). We set the ROIs for this analysis ($n = 19$) based on the brain sites that showed differential BOLD response in the es-fMRI results (Fig. 3). We constructed partial correlation matrices from mean residual time series within each ROIs for each es-fMRI run (Fig. 7a). We first compared the partial correlations within the stimulation ON versus OFF periods from all amygdala stimulation es-fMRI runs. Positive partial correlations were dominant compared to negative (anti-) correlations. Results of partial correlation connectivity between ON and OFF periods were highly similar in both positive and negative correlation connectivity (Pearson correlation $r = 0.95$ and $0.70$ for positive and negative correlation, respectively) (Fig. 7b and Supplementary Fig. 5). Leiden community detection algorithm[53] identified three parcels in the positive correlation network and they were organized as "frontal + insula + thalamus", "posterior-medial cortex + SPL", and "parietal and temporal cortex" group. Note the ACC−frontal lobe connectivity (ACC−MFG, ACC−SFG) found here were also present in the electrophysiological data (es-TT) presented in Fig. 6. This present es-fMRI analysis additionally found a significant ACC−OFC connectivity that was not evident in the es-TT data.

Next, we examined whether connectivity differed between medial and lateral group stimulation conditions during the es-fMRI stimulation ON period. The thresholded partial correlation matrix ($t$-test, FDR < 0.05) showed increased connectivity with medial amygdala group stimulation compared to lateral amygdala group stimulation in the following ROIs: ACC−MTG, PreCG−PostCG, and Insula−PCC. Note that the increased second-order ACC−temporal lobe connectivity found here was also found in the es-TT data (Fig. 6 and Supplementary Fig. 7). Increased connectivity with lateral amygdala group stimulation was found among the following connections: PCC−MTG, SPL−MCC, Insula−Precuneus, AG/SMG−STG and SFG−Precuneus (Fig. 7c). A summary of the connectivity that differed significantly when induced by medial or lateral amygdala group stimulation is shown in Fig. 7d.

## Discussion

The amygdala is thought to be a key structure subserving cognitive and autonomic functions, yet characterizing its in vivo effective connectivity while resolving major anatomical subdivisions has been difficult in humans. This information, at high spatial (subnuclear delineation) as well as temporal resolution (timescale of milliseconds) is especially important to understand network dynamics involving the amygdala. For example, determining the causal order of information flow (such as in the classical model of Parkinson's disease, as an example) in the network involving the amygdala will be important to

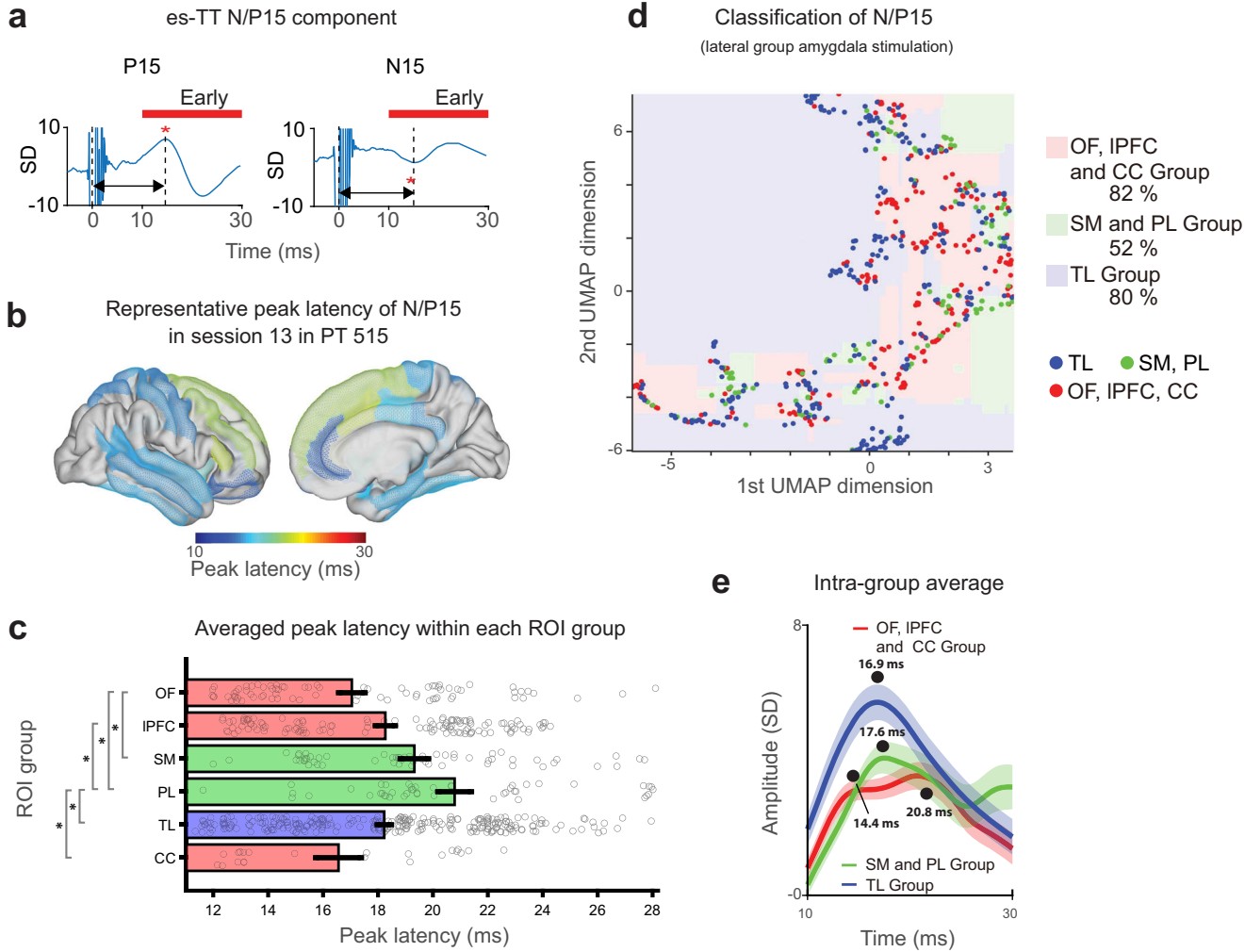

**Fig. 4 | es-TT latency analysis for early time window (N/P15) (lateral group amygdala stimulation). a** Definition and measurement for the early EPs. Red asterisks indicate peak of N/P 15 component in the EPs. Red horizontal bars indicate early time window. **b** Representative N/P15 latency distribution color-coded within ROIs on MNI brain in one patient (lateral amygdala stimulation). **c** Latency distribution within 6 anatomical groups (see "Methods"). Mean and SE (bars and error bars) are shown together with each data points represented by gray circles. One-way ANOVA showed significant difference among groups, F(5,587) = 6.93, $P = 2.65 \times 10^{-6}$. OF: orbitofrontal cortex group, lPFC: lateral prefrontal cortex group, SM: sensorimotor cortex group, PL: Parietal cortex group, TL: Lateral temporal cortex group and CC: Cingulate cortex group (anterior and dorsal posterior cingulate gyrus). Significant post hoc comparisons were indicated by asterisks on the vertical gray bars (Tukey's test, see main tex for exact P-values). Number of EPs are

77, 120, 58, 46, 270, and 22 for OF, lPFC, SM, PL, TL, and CC group, respectively. Color of bars indicates ROI groupings used in the classification analysis used in (**d**). See also Supplementary Fig. 3a for medial amygdala stimulation. **d** Early component (N/P15) classification with UMAP followed by CatBoost classifier. Classification boundary is shown as color-coded areas in 2-dimensional UMAP plane. Each colored dots represent intracranial electrode contacts. Classification accuracy is also shown in percent. **e** Intra-group averaged potentials. Here, the average is taken within the each 3 parcellations used in the analysis shown in (**d**). Black dots represent N/P15 peak latency and their values for each group. Lines and shadings around them represent mean and SE. For the creation of brain background in panel (**b**), we used Conte69.32k surface mesh[103] obtained from https://biomedia.doc.ic.ac.uk/brain-parcellationsurvey/) under WU-Minn HCP Consortium Open Access Data Use Terms.

better understand psychiatric disease, just as it is in movement disorders. The present study investigated effective connectivity from medial and lateral amygdala subdivisions with direct electrical stimulation in epilepsy patients by means of concurrent measurements of iEEG (es-TT) or BOLD signals (es-fMRI). Our results showed: (1) Electrical stimulation of the medial and lateral amygdala subdivision evoked widespread response in the human brain at two prominent latencies centered at 15 and 150 ms. The spatial distribution of these evoked potentials differed between the medial and lateral subdivision stimulation conditions: The difference was significant in OFC, ACC, pre- and post-central gyrus, STG, MTG, superior temporal sulcus, FG, and AG. Concurrent measurement of BOLD signal with electrical stimulation showed similar patterns in a largely independent patient population. (2) Analyses of the latency distribution pattern of N/P15 showed that the earliest stimulation-induced

responses were in ACC and OFC, followed by lateral prefrontal cortex and lateral temporal cortex, and finally parietal cortex. (3) The P150 latency distribution patterns also showed the OFC and ACC as having the shortest latencies for this evoked potential component. (4) Effective connectivity analysis of es-TT data suggested ACC, lPFC (SFG and MFG), OFC, and lateral temporal cortex subserve bidirectional information flow after medial group stimulation. Notably, the ACC → lPFC connectivity showed a marked difference between medial and lateral group stimulation (from conditional Granger causality analysis). Functional connectivity from es-fMRI data revealed stable intrinsic connectivity across state (es-ON and OFF) with significant partial correlations between ACC, prefrontal cortex (SFG, MFG, and IFG), OFC, and insula. The ACC–lPFC connectivity found with es-TT was also present in the es-fMRI. Finally, distinct patterns of second-order connectivity arising from stimulation of

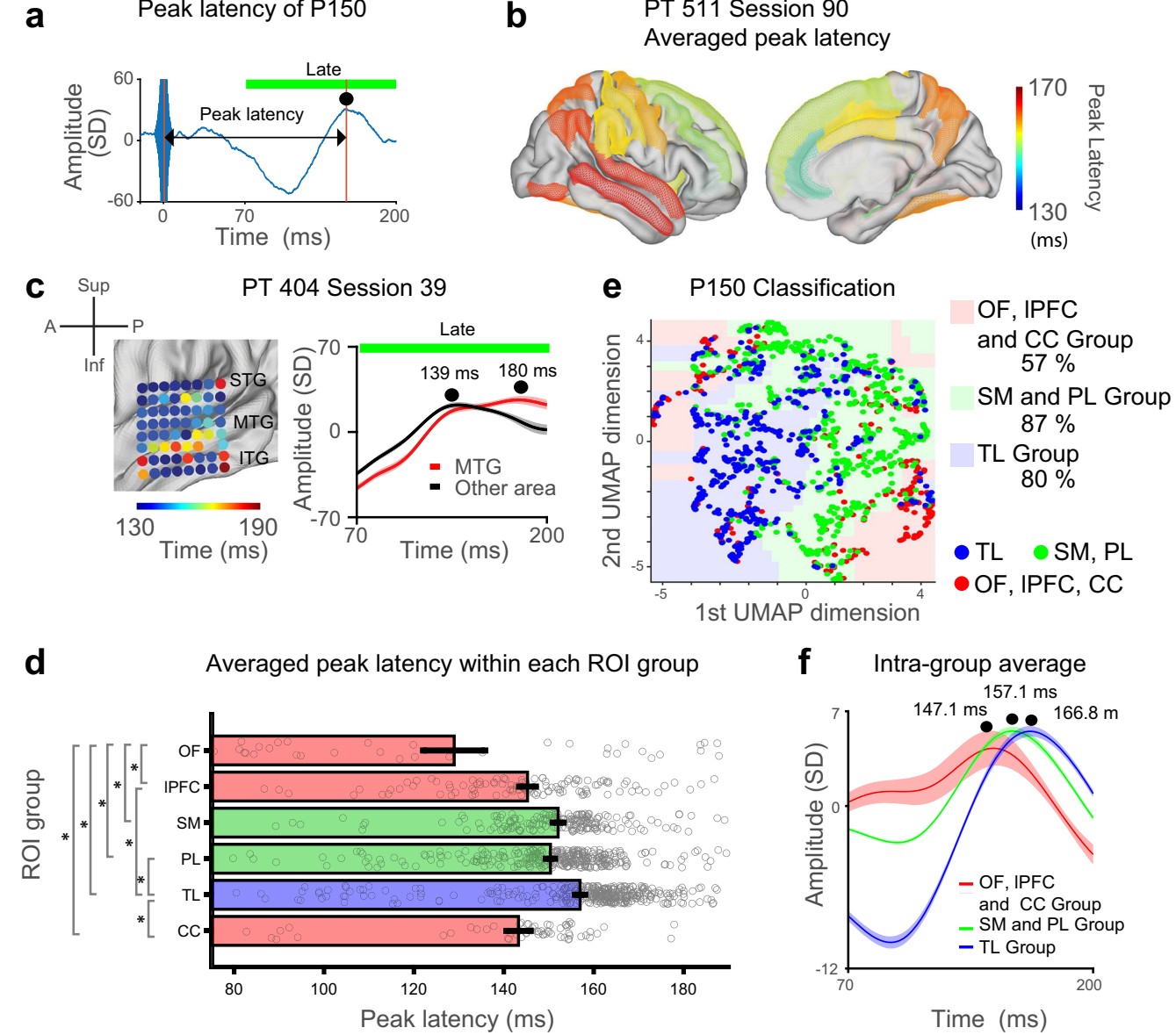

**Fig. 5 | es-TT latency analysis for late time window (P150) in medial amygdala stimulation. a** Definition and measurement of peak latency of P150 component in the EPs. Green horizontal bars indicate late time window. Black dot indicates peak of P150 component. **b** Peak latency distribution in one patient (PT511) plotted on the MNI brain. Color represents peak latency. **c** Peak latency distribution in another patient obtained with high-density grid. Right panel shows ROI averaged EPs time course. Black dots represent peak latency for MTG and other anatomical sites covered by the grid. Lines and shadings around them represent mean and SE. **d** Latency distribution within 6 anatomical groups (same as in Fig. 4c). Mean and SE (bars and error bars) are shown together with each data points represented by gray circles. One-way ANOVA showed significant difference among groups, $F_{(5,1032)} = 20.17$, $P = 3.08 \times 10^{-19}$. Significant post hoc comparisons were indicated by asterisks on the vertical gray bars (Tukey's tests, see main text for exact $P$-values). Number of EPs are 34, 110, 179, 310, 345, and 60 for OF, lPFC, SM, PL, TL,

and CC group, respectively. Abbreviation is the same as in Fig. 4. See Supplementary Fig. 3b for lateral amygdala stimulation. Color of bars indicates ROI groupings used in the classification analysis in (**e**). **e** Late component (P150) classification with UMAP dimensional reduction and CatBoost classifier. Classification boundary is shown as color-coded areas in 2-dimensional UMAP plane. Each colored dot represents intracranial electrode contact. Classification accuracy is also shown in percent. **f** Intra-group averaged potentials. Here, the average is taken within the each 3 parcellations used in the analysis shown in (**e**). Black dots represent P150 peak latency and their values for each group. Lines and shadings around them represent mean and SE. For the creation of brain backgrounds in panel (**b**), we used Conte69.32k surface mesh[103] obtained from https://biomedia.doc.ic.ac.uk/brain-parcellationsurvey/) under WU-Minn HCP Consortium Open Access Data Use Terms.

medial vs lateral amygdala group involved ACC, MCC, PCC, sensorimotor cortex, and temporal lobe.

The spatial distribution of responses we observed with both es-TT and es-fMRI is largely congruent with prior reports in experimental animals, notably amygdala connections with orbitofrontal cortex, anterior cingulate cortex, and insula, which subserve critical cognitive functions. The orbitofrontal cortex is important for decision-making and reward contingency[54–56], the anterior

cingulate cortex is implicated in depression, pain, and emotion[57,58], and the insula is involved in interoceptive processing[59,60], autonomic control, and again depression[61]. Additional connectivity was found between the amygdala and sensorimotor cortex, a novel finding in humans (for related findings in animal studies, see refs. 62–65). There was a largely congruent distribution of the differential response (between medial vs lateral amygdala group stimulation) across many of these regions when comparing es-TT and

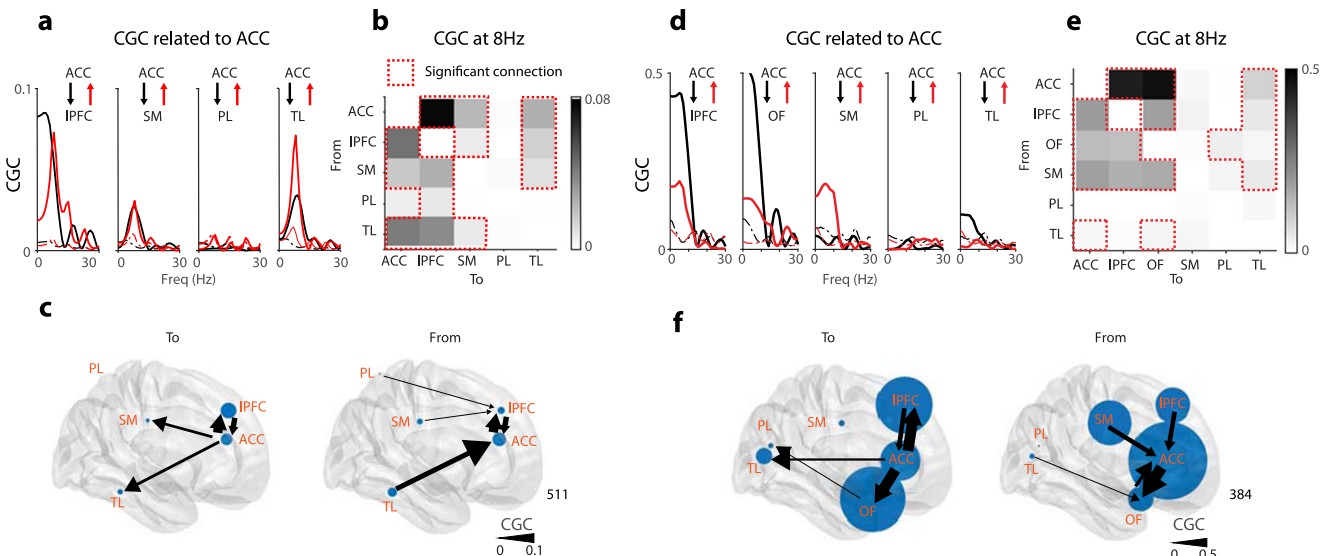

**Fig. 6 | Results of Conditional Granger causality of intracranial field potentials in two patients. a** Averaged spectral Conditional Granger causality (CGC) related to anterior cingulate cortex (ACC) across 12 sessions of esTT experiment in PT511. x-axis indicates iEEG frequency in Hz and y-axis indicates magnitude of CGC. Solid lines represent CGC calculated from actual data and dotted lines represent 95% CGC values found with phase-randomized surrogate data. **b** Averaged CGC among 5 ROI groups. Peak CGC values at 8 Hz were shown. Colorbar represents CGC values at 8 Hz. The cells in the matrix surrounded by the red dotted lines indicates significant CGC (over 95% point from phase-randomized surrogate data distribution). **c** Effective connectivity (CGC) plotted on the MNI brain among ROIs. Thickness of the arrows are proportional to the CGC values of the connections (scale at right lower corner of the panel), and size of the circles represent sum of the inflow (left panel) to and outflow (right panel) from the ROIs (in-degrees and out-degrees, respectively). Location of the circles reflects actual electrode placement in the patient. **d**, **e**, **f** Same as (**a**), (**b**), and (**c**) above, but for PT384. Note PT384 had ROI coverage in OF group. Both were from medial group amygdala stimulation (See Supplementary Fig. 7 for lateral group amygdala stimulation). For the creation of brain backgrounds in panels (**c**) and (**f**), we used Conte69.32k surface mesh[103] obtained from https://biomedia.doc.ic.ac.uk/brain-parcellationsurvey/) under WU-Minn HCP Consortium Open Access Data Use Terms.

es-fMRI, despite differential brain coverage for these two dependent measures.

Our study also provides important data on the timing of responses on the millisecond scale. The earliest potentials in response to the amygdala stimulation had latencies around 15 ms (N/P15). With lateral group stimulation, the N/P15 component showed a clear graded pattern of latency reflecting the timing of neural responses: the pattern was such that responses were observed first in the anterior cingulate and orbitofrontal cortex (OF and CC group), followed by in the lateral prefrontal and lateral temporal cortex (lPFC and TL group). And the parietal cortex (PL group) showed the slowest response within the ROI groups. The later P150 component had a very similar propagation order. Analysis of both EP components indicated the anterior cingulate cortex and orbitofrontal cortex have significant, short-latency effective connectivity from the lateral part of amygdala, while sensorimotor cortex and parietal lobe respond with longer latencies. A similar propagation pattern was observed with medial group stimulation, except that somewhat longer latencies were seen in anterior cingulate cortex. Temporal order reflected in EP latencies among ROIs was largely comparable (Figs. 4c, 5d and Supplementary Fig. 3) between N/P15 and P150 components, which may be attributable to dependencies between N/P15 and P150 generation. N/P15 likely reflects the excitatory response of pyramidal cells at sites that have direct anatomical connections with the amygdala, whereas P150 is thought to reflect local processing (e.g., sustained suppression) triggered by either the fast local excitation response (reflected in the N/P15)[66–68] or by multisynaptic functional connection.

To quantify second-order (network-level) connectivity from es-TT data, we applied conditional Granger causality analysis to discover information flow between the nodes in the co-activation patterns in the iEEG signals. Specifically, ACC, lPFC, and lateral temporal lobe were the key components carrying the second-order information flow. For the BOLD-fMRI data, we applied residual time series partial correlation[69,70] and obtained remarkably similar findings. In line with prior studies in humans[70–72], that found that brain's network architecture is dominated by stable "intrinsic" networks within which task-related changes are relatively small, we found that ACC–OFC connectivity was significant in both es-ON and es-OFF periods. Together with the results from the latency analyses provided by es-TT (ACC and OFC had the earliest latencies), we suggest that ACC and OFC are important nodes that influence downstream regions in the amygdala network.

The second-order (network-level) connectivity we found between ACC and lateral temporal cortex (STG and MTG) is intriguing in light of large anatomical differences between humans and monkeys in lateral aspects of the temporal lobe: monkeys lack a distinct MTG, whereas MTG in humans is well developed and distinct. Tracer studies in monkeys have demonstrated that the cingulate cortex projects directly to the amygdala[73–75] and that the amygdala projects to the temporal lobe including area TE (a higher-order visual area just inferior to the STS) in monkeys[76]. The amygdala might thus relay signals from ACC to STG and MTG.

The preliminary findings on hemispheric laterality and sex that we presented will be important to pursue in future studies with larger samples, and for studies in personalized medicine of psychiatric disorders. Our findings support the idea of previously reported female-left/male-right interaction patterns[50,77] and extend prior reports based on activation within the amygdala to effective connectivity from amygdala to widespread brain areas. In light of apparent unequal sex distributions in many psychiatric illnesses such as depression and anxiety, it will be critical to include sex as a key factor in future studies that might apply our approach to questions in psychiatric illness.

Our results also suggested possible associations between depression or anxiety in our subjects and amygdala connectivity with prefrontal cortex, lateral temporal cortex, AG, and insula. It is worth mentioning that these effects were prevalent for later P150 components that likely reflects functional (as opposed to structural) connectivity, and showed area-specific hyper- and hypo-connectivity.

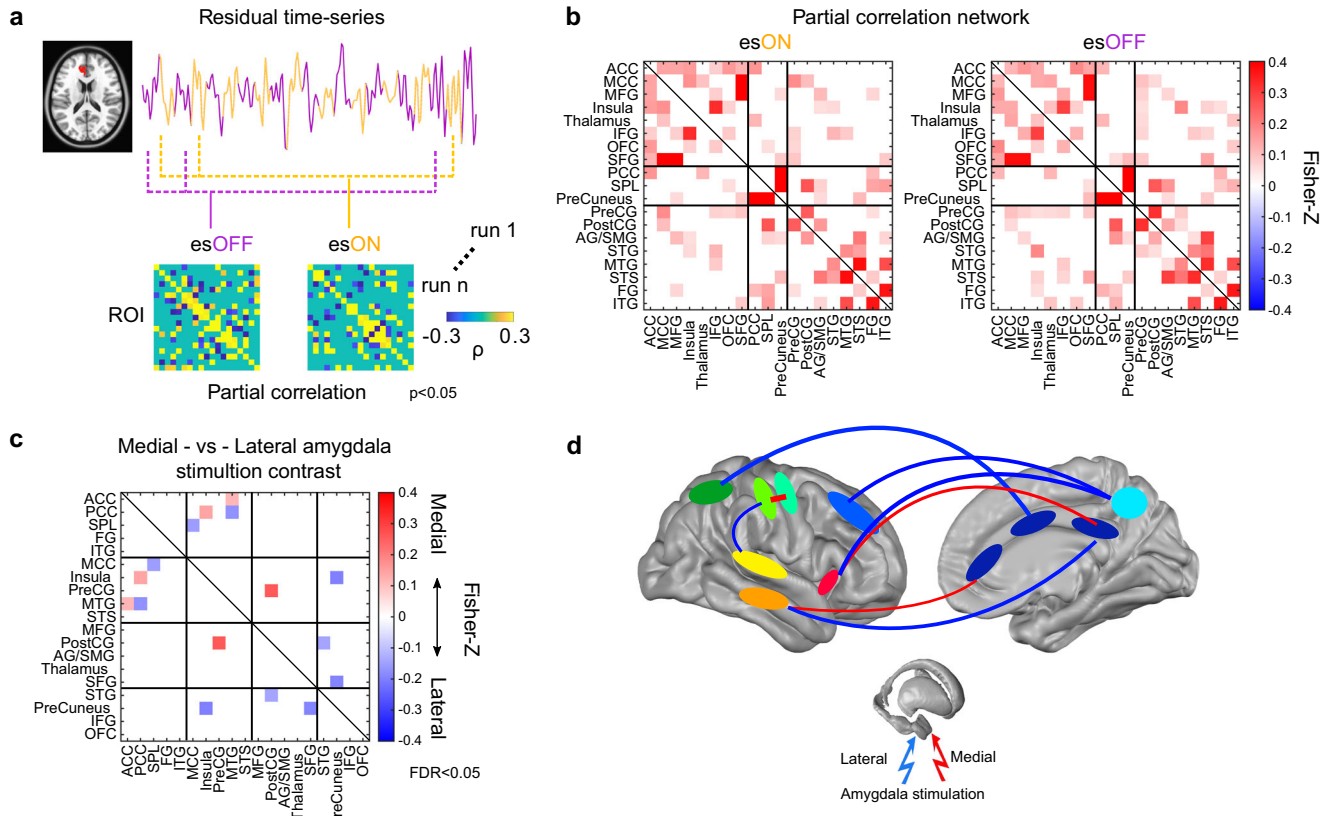

**Fig. 7 | Second-order connectivity analysis on the es-fMRI residual time series.**
**a** Procedure to compute the connectivity within electrical stimulation (es-) ON and OFF period. Red patch on the MRI image shows the location of a ROI for example. Average residual time series within the ROI is shown in yellow (esON) and purple (esOFF) time series plotted on the right. Example partial correlations ($\rho$) among ROIs were shown below separately for es-ON and es-OFF period (thresholded at $P < 0.05$, two-sided t-test, uncorrected). **b** Overall averaged partial correlation networks were shown for es-ON and es-OFF period. Color indicates Fisher-Z values. The community parcellation (indicated with black lines) was derived with Leiden algorithm. For negative correlation network, see Supplementary Fig. 5. **c** Contrast of partial correlation in the es-ON period between medial and lateral amygdala subdivision stimulation (thresholded at FDR < 0.05, two-sided t-test). Hot and cold colors indicate connection is larger for medial amygdala stimulation and vice versa.

**d** Schematic representation of BOLD connectivity significantly different between medial and lateral amygdala stimulation. Linewidth is proportional to the magnitude of difference of connectivity (in Fisher-Z values). Red and blue lines indicate larger partial correlation connectivity for medial amygdala than lateral amygdala stimulation and vice versa during es-ON period. For the creation of brain backgrounds in panels (**d**), we used Conte69.32k surface mesh[103] obtained from https://biomedia.doc.ic.ac.uk/brain-parcellationsurvey/) under WU-Minn HCP Consortium Open Access Data Use Terms. For the creation of brain background in panel (**a**), we used template ICBM152 Non-linear brain obtained from http://nist.mni.mcgill.ca/icbm-152-nonlinear-atlases-2009/ under Copyright (C) 1993–2004 Louis Collins, McConnell Brain Imaging Centre, Montreal Neurological Institute, McGill University.

However, these exploratory analyses were not pre-planned and severely limited by small sample size.

While offering unique data in humans, our study is limited in a number of important respects. First, it is possible that aspects of amygdala connectivity are reorganized in patients with long-standing epilepsy, as is known to happen in some other cases[78,79]. It is possible that our results might not generalize to healthy populations straightforwardly, given the relatively small sample size with various clinical conditions in our subjects. For example, one could further ask, within the epilepsy population, whether finer subgroupings could be made depending on various factors, such as epilepsy with normal MRI finding or long-standing history of epilepsy. A considerably larger sample size would be needed to answer these and other questions about individual differences. Second, the electrode coverage differed among our subjects due to clinical considerations. There may be a number of structures that were not covered by the current study, or that were undersampled with es-TT (e.g., posterior cingulate). To ameliorate this issue, we combined the whole-brain field-of-view provided by es-fMRI with the data from es-TT in the current study so that brain structures not covered with intracranial electrodes could be sampled with the neuroimaging technique. Third, with inevitable current spread and the relatively small size of

the amygdala complex in humans, stimulation may not have been confined entirely within the medial and lateral amygdala groups as intended. This possible confound was ameliorated by using bipolar electrical stimulation with adjacent contacts to minimize current spread. Since clinical macro-electrodes were used for stimulation and recordings, we did not attempt to quantify indirect effects of our stimulation that might involve circuits intrinsic to the amygdala. The internal circuitry of the amygdala is typically investigated with circuit-level resolution, using techniques such as optogenetics, that are not yet possible in humans[80,81], or microstimulation–microrecordings. These will be important future extensions to complete the brain-wide level of analysis provided in our present paper. Finally, despite the strength of high temporal resolution in our study, reciprocal connections between the amygdala and its targets, as well as the state dependency of those connections[82], further complicates any simplistic interpretation of the results. Future studies, including further work in animals, will be required to provide convergent evidence with the present studies. Despite these limitations, we offer an unprecedented dataset that will inform neuroimaging studies of the human amygdala, and that can serve to leverage network-level studies of the amygdala's function in healthy individuals as well as in psychiatric illness.

## Methods

### Statement of ethics

All experimental protocols were approved by the University of Iowa and the California Institute of Technology Institutional Review Boards. All participants provided written informed consent for this study. For the subjects under 18 years old, written informed consents were obtained from parents.

### Subjects

In total, 25 neurosurgical patients (17 males, 8 females; mean and standard deviation of age, $28.1 \pm 13.3$ years) were recruited for this study. The participant demographics are shown in Supplementary Table 1. These participants were selected from 46 consecutive patients who underwent intracranial electrode implantation for chronic monitoring of their epilepsy after August 2016. From this pool of patients, all subjects with stimulus sites in the Amygdala, confirmed by postoperative MRI images, were included. For the es-TT experiment, 13 patients were included in this study (mean age = 26.3 years old, 6 female and 7 male) and for the es-fMRI experiment, 16 patients were included (mean age = 29.4 years old, 2 female and 14 male). Four patients underwent both procedures.

### Intracranial electrodes and recording

Electrodes implantation plans were tailored to the clinical needs of each patient (e.g., suspected seizure focus, potentially involved surrounding areas) and based on preoperative evaluation by the multidisciplinary epilepsy team. The location and number of electrodes varied depending on these clinical considerations. Before electrical stimulation, each participant's anti-seizure medications were restarted. Depth electrode arrays (4 to 10 platinum macro-electrode contacts/depth electrode shaft, 1.3 mm diameter, 1.6 mm length, spaced 5–10 mm apart, Ad-Tech Medical, Racine, WI) were stereotactically implanted in each subject (Fig. 1a). Subdural grid arrays consisted of platinum-iridium disc electrodes (2.3 mm exposed diameter, 5–10 mm inter-electrode distance) embedded in a silicon membrane. In situ impedance measurements were made, and they were 1.5–5.0 kOhm at 100 Hz for both depth and grid contacts. Electrocorticographic (ECoG) and stereo electroencephalographic (SEEG) data acquisition was simultaneously made by multichannel data acquisition system (ATLAS system using Pegasus software version 2.2 2., Neuralynx, Tucson, AZ) with a reference contacts placed in the patients' subgaleal space. The sampling rate was 8000 Hz with 1–2000 Hz acquisition filters (−6 dB, 256 tap length).

### Electrode localization and amygdala subnuclei parcellation

Location of the electrode contacts was determined using pre- and post-electrode implantation MRI and CT imaging. For each participant, volumetric high-resolution T1-weighted structural 3 T MRI scans were obtained before and after electrode implantation. Pre-implantation scans were obtained with GE Discovery 750 W scanner with a 32-channel head coil. T1W inversion recovery fast spoiled gradient recalled (BRAVO) sequence, $1.0 \times 1.0 \times 0.8$ mm voxel size, TE = 3.28 ms, TR = 8.49 ms, TI = 450 ms, FOV = 240 mm, flip angle = 12 degrees). Post-implantation MR scans were obtained with Siemens Skyra scanner with a head transmit-receive coil (MPRAGE sequence with $1.0 \times 1.0 \times 1.0$ mm resolution, TE = 3.44 ms, TR = 1970 ms, TI = 1000 ms, Flip angle = 10 deg, FOV = 250 mm). We also obtained volumetric thin-slice post-implantation CT scans with 1.0 mm slice thickness. The post-implantation CT and MR structural scans were first linearly co-registered to pre-implantation MR scans using FLIRT module of FSL. Images were corrected for displacement and deformation from surgery using non-linear three-dimensional thin-plate spline warping using manually selected control points. Location of the electrode contacts was identified with post-implantation MRI and CT scans and transferred onto the pre-implantation participant's MRI space and template brain (MNI) space. FreeSurfer (version 7.2.0) was used to create cortical surface mesh, segmentation, and parcellation of brain using the pre-implantation structural MR images[83].

Amygdala parcellation used in this study came from the CIT168 human brain template built with 168 healthy human subjects through highly accurate non-linear registration of T1- and T2-weighted MR images that were co-registered onto MNI space[33,84]. Structural relationships between electrode contacts and subnuclei of the amygdala were determined from the projected contact locations (MNI coordinates) onto the CIT168 template which was further transformed in the MNI space (Figs. 1d and 3a).

Majority of the stimulation sites in this series fell within the lateral nucleus as well as in the dorsal and intermediate part of basolateral nucleus of the amygdala (lateral, basolateral nucleus in Mai's and Allen's human brain atlas). We set a boundary line separating the medial and lateral group of amygdala sites using the lateral nucleus (La) parcellation (Figs. 1d and 3a). If medial contact of the pair was located medial to the border between lateral nucleus, that stimulation location was regarded as medial group amygdala stimulation. We note that there was no stimulation site in the central nucleus, and we have paid careful attention not to include contacts in the hippocampus and amygdalo-striatal transition area (ASTA). In summary, our boundary separates stimulation sites into two parcels, the lateral group parcel includes lateral nucleus of the amygdala, and the medial group parcel includes basolateral and accessory basal nucleus of the amygdala. Note that in Mai's[85] and Allen brain atlas (http://atlas.brain-map.org/), accessory basal nucleus is refereed as basomedial nucleus. Each dot in Fig. 1d and Fig. 3a represents the midpoint of pairs of stimulation electrodes.

### Electrical stimulation tract-tracing (es-TT) procedure

Electrical stimulation–tract tracing (es-TT) uses direct and focal electrical brain stimulation together with concurrent intracranial electroencephalography (iEEG) recordings from other brain sites obtained from other depth electrodes (stereotactic electro-encephalography: SEEG) as well as from subdural strips and grids on the surface of cortex (electrocorticography: ECoG) to provide direct evidence for specific connectivity with high temporal resolution[86–88].

For the series, we analyzed 51 amygdala stimulation es-TT runs (30 runs from medial group and 21 runs from lateral group stimulation). Some amygdala sites were stimulated multiple times. In total, 37 sites (17 in the left and 20 in the right) in the amygdala were stimulated (17 lateral, 20 medial; mean 2.85 sites per participant, 1–6 amygdala sites/patient; Supplementary Table 1). Stimulation waveforms were controlled by a real-time processor (RZ-2, Tucker-Davis Technologies, Alachua, FL). Stimulus parameters were chosen based on three factors: safety, efficacy of neural stimulation, and minimizing artifacts. Single-pulse electrical stimulation was applied in a bipolar fashion using a pair of adjacent electrodes in the amygdala. Sixty charge-balanced biphasic square-wave electrical pulses (0.2 ms duration per phase, 0.4 ms total duration) of alternating polarity were delivered at 0.5 Hz with small temporal jitters. Stimulation was current controlled using linear analog stimulus isolators (AM Systems, model 2200) with current intensity of 9 mA (Fig. 1a). The stimulus intensity was chosen based on empirical safety limits and the prior reports[87–89]. Calculated charge density and charge/phase were ~22.6 $\mu C/cm^2$ and 1.8 $\mu C$/phase, respectively, and these values falls within the safety limit, at the same time, can achieve high-enough current stimulation for neural tissue[90–92]. The pulse width of our stimulation was very close to the chronaxie measured in human and monkey brain indicating the stimulus duration was long enough for neural tissue excitation[93,94]. Our stimulation frequency was very slow (0.5 Hz) and brief (total duration was 0.4 ms). This allows us to study short latency waveforms and avoid local charge accumulation. Continuous iEEG was recorded using the ATLAS system (NeuraLynx) with a sampling frequency of 8 kHz with 2 kHz anti-aliasing filter. The

stimulation was delivered while the participants were resting on their bed in the electrically shielded epilepsy monitoring room. We analyzed 11956 EPs in total. Contacts localized within the white matter were excluded, and EPs that shows signs of amplifier saturation and large non-physiological fluctuation (i.e., cable movement) were also discarded. Resulted 3720 EPs were retained for further analysis. Overall, we analyzed es-TT responses from 1447 recording sites over 13 subjects (712 subdural surface contacts and 735 depth electrode contacts) (Fig. 1b, c). Although some patients had bilateral amygdala depth electrodes (6 out of 13 patients), permitting stimulation of either left or right amygdala, electrode coverage was generally unilateral. A prior report of electrical stimulation of the human amygdala showed that the response in the limbic system was observed exclusively in the ipsilateral hemisphere[90]. Another tracer injection study in NHP also indicated that direct inter-hemispheric connection of amygdala was extremely sparse[91]. Further, a critical point is that inclusion of hemisphere contra-lateral to the stimulation adds confounds regarding the latency analyses. For these reasons, we restricted all analyses to stimulation-recordings within the same hemisphere. Results from left and from right hemisphere are pooled for the initial analysis, then we divided the dataset into right and left hemisphere.

## Electrical stimulation–functional MRI (es-fMRI) procedure

We also employed concurrent intracranial electrical stimulation and functional MRI on 16 neurosurgical patients to map the effective connectivity from amygdala. This is particularly useful to examine the connectivity between the stimulation site and other brain areas where no electrode was placed (electrophysiological data cannot be obtained from these areas).

Details of the es-fMRI procedure were also described elsewhere[31,88]. The subjects were scanned in resting condition (no task was delivered) on the day prior to electrode removal surgery. Blood-oxygenation-level-dependent (BOLD) images were obtained concurrently during intracranial electrical stimulation with Siemens Skyra 3 Tesla MR scanner (NUMARIS, Syngo MR E11). Gradient-echo echo-planar imaging (EPI) was used with following parameters to obtain T2* images (TR = 3.0 s, TE = 30 ms, slice thickness = 3.0 mm, FOV = 220 mm, Flip angle = 90 degrees, phase encoding lines = 68). Stimulus waveform was generated in the control computer and delivered to the analog linear stimulus isolators (AM Systems, model 2200). Stimulus waveform was charge-balanced biphasic square pulses (0.25 and 0.75 ms duration) with inter-phase interval (0.25 ms). Seven to nine pulses were delivered at 100 Hz repetition rate. The fMRI utilized block design and the stimulation was repeated for 10 consecutive TRs followed by no stimulation period (30 s). We discarded the data from 6 subjects due to the stimulation site did not fall into the amygdala. Overall, we analyzed 30 amygdala stimulation es-fMRI runs over 16 subjects (13–43 years old, mean age = 29.4 years, left amygdala stimulation = 16, and right amygdala stimulation = 14).

## Quantification of the es-TT responses

The es-TT responses were analyzed by the following procedure. First, the trials that contain non-physiological signals (e.g., cable motion artifact, long decay artifact from amplifier saturation) and potential interictal spikes that showed large absolute amplitude (over 10 times standard deviations at 8 ms that was usually the end of stimulus artifacts, or over 50 times standard deviation within 8–1000 ms after stimulation, standard deviations were calculated within 100–10 ms before the stimulus for each trials) were discarded, then evoked potentials (EPs) were calculated by averaging resultant single trials channel by channel (60 trials per one stimulation run). EPs were then normalized with respect to their standard deviations (SD) within the baseline temporal window (100 to 10 ms before the onset of electrical stimulation). EPs from contacts located in white matter and seizure onset zone, or outside the brain were discarded from analyses. Each

contact was assigned to anatomical ROIs. These ROIs were determined according to Destrieux's parcellation[47]. Destrieux's parcellation has 74 segmentations of gyri and sulci for each hemisphere. Subcortical ROIs were also included (see "es-fMRI data analysis and quantification" section). Obtained EPs were further averaged within each ROI; then intra-ROI averaged EPs of medial group stimulation were compared with those of lateral group stimulation (Fig. 1e).

We quantified the magnitude of es-TT responses by calculating range values (the difference of maximum and minimum amplitude of the es-TT evoked responses within a time interval) of normalized EPs within two temporal windows, early and late (10–30 and 70–200 ms after onset of electrical stimulation, respectively, Fig. 2a) according to the methods used in the prior report[88]. Latency of the EPs was found by detecting the time after stimulus onset at which the filtered Eps waveform reached peak within the time window. The earliest peak within the early time window was defined as N15 or P15, because the peak of this early component is located at -15 ms after stimulation. When the earliest peak of intra-ROI averaged EP was positive polarity, it was defined as P15. On the other hand, the earliest peak was negative polarity, it was defined as N15 (Figs. 2b and 4a). For the calculation of the range values in the early temporal window, we applied band-pass filter (2nd-order Butterworth IIR bandpass filter with 6 dB cutoff frequency at 2 Hz and 200 Hz, no group-delay) to suppress effect of baseline drift from late response. Positive large potentials were observed in the late time period (70–200 ms) in many EPs with visual inspection (Fig. 2a and Supplementary Fig. 1). Since the peak of this late component is located at -150 ms after the stimulation with positive polarity, we defined this component as P150 (Figs. 2d and 5a). Since we were particularly interested in the latency distribution, and due to the fact that latencies could not be unambiguously determined for EPs that did not show clear peak, channels that did not show supra-threshold es-TT responses were not used for further analyses.

We set thresholds at 95 percentile points calculated from the amplitude histogram in the baseline time window (1000–10 ms before the onset of electrical stimulation). If the EPs amplitude (measured as the range value) exceeds this threshold in the post-stimulus time window (10–200 ms), that EPs were considered to be significant.

## Comparison of EPs waveform

We first examined EPs amplitude difference for each ROIs between medial and lateral group stimulation condition by two-sided t-tests on the range values with Bonferroni correction. P (corrected) <0.05 was considered as significant.

As shown in initial response mapping (Fig. 2b, d), the EPs were broadly distributed across brain and not limited to particular brain sites. To confirm the distribution discriminated the stimulus location group, we applied the machine learning analysis. We used state of the art algorithms for dimensional reduction (UMAP[49]) and pattern classification (CatBoost[48]). Both algorithms operate in non-linear fashion so this combination could be efficient and sensitive to pick up differences in the data since the form of temporal and spatial dependence can well be non-linear.

Two classification analyses were performed to examine the existence of differential es-TT response in the es-TT dataset. Initial classification analysis (Fig. 2f) was done using the quantified es-TT response values (see above section). For each intracranial contact, the range values, peak latency, ROI identity and trend values were calculated and assigned to the contact. Trend values were calculated as voltage difference between the first and last data points within the time windows. ROI IDs were treated as categorical feature (non-numeric factor) and other features were treated as continuous variables.

The second classification analysis (Fig. 2g) was done on EPs waveform in a reduced dimension along time axis with a non-linear dimensionality reduction with manifold embedding technique, Uniform Manifold Approximation and Projection: UMAP[49]. The 1520

temporal dimension (corresponding to 10–200 ms after stimulus onset) was projected onto 2–100 UMAP layout coordinates. We used the following parameters: number of neighbors = 15 and minimum distance = 0.1 with euclidean distance metrics for the UMAP dimensional reduction. We used UMAP (umap package version 0.2.7.0) implemented in R (version 4.0.2 running on Windows 10) environment. Note that in this dimensional reduction stage, neither knowledge about the spatial distribution nor EPs peak latencies and its amplitudes were needed.

For both classification analyses, Catboost (gradient boosting algorithm on classification trees[48]) was used, and the task is to predict the stimulation site (medial or lateral group stimulation that was assigned to each contact). For the Catboost classification, we used following parameters (tree-depth = 14, 200 iterations, Logloss, learning rate = 0.03). Of note, Catboost supports non-numerical categorical input features (such as ROI identity) by transforming categorical feature within its library. We used the algorithm available in R package (catboost, version 0.23.2). Performance of classification was accessed with 10-fold cross-validation. We also applied the same procedure with shuffled data obtained from exactly the same dataset. Classification accuracy is reported.

Peak latency difference among the ROIs were tested by one-way ANOVA (Figs. 4c and 5d) followed by Tukey's post hoc tests. To find overall waveform shape difference across the ROI groups, we also applied UMAP to the EPs waveform as before (Time x Channel matrix; 1040 timepoints x 1282 iEEG recording channels, corresponding to 70–200 ms from the stimulation) using the same parameters as above. The resulted 2 UMAP-dimensional projection of the waveform were subjected to Catboost classifier. We divided the EPs into three groups (Frontal group: OFC, lPFC, and CC, Parietal group: SM and PL, and Temporal group: TL). Classification boundaries were evaluated at $400 \times 400$ points in 2-UMAP dimensional plane and accuracy was found by counting the number of EPs that correctly classified and divided by the total number of EPs in each group.

Further, we have examined the effect of two factors: Sex and Hemisphere, on es-TT responses. We calculated effect size (mean amplitude) and its standard error for each ROI group and presented as errorbar graphs (Supplementary Fig. 8). Note we restricted the analyses to ipsilateral recording to the stimulated hemisphere.

We also examined differences in es-TT response between patients with and without depression and/or anxiety. We used neuropsychological evaluations done before electrode implantation surgeries. The evaluations were done by clinical neuropsychologists. es-TT response amplitudes were pooled within each anatomical ROI and t-tests were performed ROI by ROI. P-values <0.05 with Bonferroni correction were considered significant.

## The effective connectivity analysis of es-TT data

We examined the effective connectivity among the ROIs with non-parametric spectral conditional Granger causality (CGC) analysis[95,96] in three of the patients (384, 511, and 515) since these patients have multiple es-TT runs for robust calculations of CGC while their electrode coverage included all structures of interests. This non-parametric spectral approach developed by Dhamala and colleagues[97] was used to avoid multivariate autoregressive (MVAR) model misspecification. Importantly, the method can yield directional connectivity conditional on all variables, and the result from CGC could be regarded as direct influence between variables (channels) after all the indirect influences (for example, effect of common source) are accounted for. All pairs of spectral CGC were calculated for each medial group stimulation session and resulted spectral CGCs were averaged over the es-TT runs. Significance of CGCs were evaluated using phase-ramdomized surrogate data. The phase of the original LFPs data was randomly rearranged independently for each contact, which destroys the systematic causal

relationship between contacts leaving only chance occurrence while keeping the power spectrum (=autocorrelation) the same as original data. Two hundred times of iteration were performed for each session. Since CGC values usually peaked around 8 Hz, the CGC values that exceeded the threshold (95% point from 200 iterations averaged over 12 es-TT runs in the surrogate datasets) at 8 Hz were considered to be significant.

## es-fMRI data analysis and quantification

All imaging data including anatomical and es-fMRI data were curated according to the BIDS standards[98]. The imaging data were pre-processed using fMRIPrep pipeline[99]. The pre-processed data were available in OpenNeuro[100]. Briefly, the pipeline includes the following processes. Intensity correction, estimation of brain mask and spatial normalization to the ICBM template (ICBM152, Non-linear asymmetrical template) of T1w anatomical images. Slice timing correction, motion correction, distortion correction with field map, coregistration of the subject's T1w images to the MNI template, and coregistration of BOLD images to the template were performed. Frame-wise displacement (FD) was also calculated.

First-level analyses of the es-fMRI data were done with deconvolution with AFNI's 3dDeconvolve with 14 parameter (−3 to 36 s from onset of stimulation block) CSPLIN (cubic spline) function after normalization of the signal (that makes the mean signal intensity to 100). The procedure performs variable shape regression just like general linear model (GLM) fitting with finite impulse response (FIR) basis sets and allows the shape of BOLD response to vary for each voxel. The es-fMRI image frame that showed large motion (FD > 0.9 mm) and one frame before that frame were discarded (censored). Spatial smoothing with gaussian kernel (full width at half maximum = 6.0 mm) was applied before deconvolution. 12 motion regressors (6 motion regressors and their temporal derivatives) and top 6 aCompCorr components were included as noise regressors (in total, 18 noise regressors were included). We further employed generalized least squares fit using AFNI's 3dREMLfit to take temporal autocorrelation into account by estimating ARMA (autoregressive moving average) model parameters. Results from this generalized least square fit were used in this paper. Datasets showing evidence of a response anywhere within the brain mask (false-discovery rate corrected Z > 2) were submitted to higher-level analysis (30 es-fMRI runs). Resulting statistical maps were subjected to multi-level mixed-effects analysis using AFNI's 3dMEMA for group-level analysis. This takes the precision information of beta coefficients estimation into account. The first- and higher-level GLM were conducted in standard space (MNI-152-NonLinear-2009c Asymmetrical).

For the multiple comparison correction, spatial autocorrelation of residual BOLD time series for each run was calculated using AFNI's 3dFWHMx. Trimmed mean value (upper and lower 5% were not used for the calculation of mean values) of the parameters that defines the shape of spatial smoothness across all es-fMRI runs were used to estimate cluster size threshold using AFNI's 3dClustSim for statistical thresholding of second-level analysis. We used voxel-wise threshold of P = 0.01 and cluster-wise alpha = 0.02 for finding cluster size threshold.

For the ROI-based analysis, we used the same cortical and sub-cortical parcellation as used in es-TT data analysis. The cortical parcellation was from Destrieux atlas and subcortical segmentation was from Freesurfer's segmentation stream run on the template brain[101]. Mean beta coefficients (over 3 to 30 s from onset of ON stimulation period) were extracted for each ROI.

We also employed classification analysis to examine whether individual es-fMRI runs can be reliably classified based on the ROI averaged BOLD responses between medial and lateral group stimulation. Beta coefficients obtained from 3 to 30 s after onset of stimulation ON period were averaged within each ROIs. The area that showed

signal dropout due to the intracranial electrodes was set to not-a-number (NaN). Since there was large difference in numbers of runs for medial and lateral group stimulation (imbalanced data with a ratio of 2:1, 20 and 10 runs for medial and lateral group, respectively), we applied SMOTE (synthetic minority over-sampling technique) before applying the classifier[102]. Random forest classification algorithm with 250 trees was used and its classification performance was evaluated with the out-of-bag area-under-the-curve metrics (AUC). The AUC values were evaluated with original and shuffled data (medial/lateral label was randomly shuffled 100 times). Out-of-bag variable importance for the classification was also obtained for each permutation and its mean importance values were presented. The variable importance values that exceeded the threshold calculated from the shuffled classification analysis (mean + 5 times the standard deviation) were considered to be significant.

### The second-order connectivity analysis of es-fMRI data

In addition to the deconvolution (GLM with FIR) analysis to examine the "first-order" effect of electrical stimulation as explained above, we also performed the "second-order" connectivity analysis using the es-fMRI residual time series among the selected ROIs, that is, the connectivity profile calculated from the covariance structures using time-point to time-point BOLD time series fluctuations. The residual time series were obtained by subtracting full-model (from the GLM with FIR above) fitted time series from original (but pre-processed) BOLD time series for each voxel. This residual time series have functional connectivity information (also called background connectivity, task-state connectivity or noise correlation) that was not captured by the stimulus-triggered BOLD response found by the GLM and reflects the second-order connectivity. We note that use of FIR regression is important to eliminate spurious inflation of the connectivity measure[70]. Mean residual time series within the ROI ($N$ = 19, OFC, ACC, MCC, PCC, Precuneus, Thalamus, SFG, MFG, IFG, AG/SMG, PreCG, PostCG, STG, MTG, ITG, STS, Fusiform gyrus (FG), SPL and Insula) were extracted (we used AFNI's 3dNetCorr program) and a bandpass filter was applied (0.1–1 Hz). Partial correlation among the ROIs (further transformed with fisher-z transformation) were calculated using each ROI's mean residual time series. The resulting partial correlations ($\rho$) were converted to Fisher-z scores. We applied this procedure for electrical stimulus ON period and OFF period separately for each es-fMRI runs. First 2 frames in each es-fMR runs were discarded. Community structures of partial correlation network within the considered ROIs were found using Leiden algorithm[53], implemented in leidenAlg and igraph package in R for the visualization used in Fig. 7 and Supplementary Fig. 5. For this, we set the resolution parameter to 1.0 (=default value). We performed consensus clustering on the resulted Leiden's membership to get the final parcellations ($N$ = 200). For the comparison between electrical stimulation ON versus OFF condition, $t$-tests were performed for each ROI pairs. False-discovery rate (FDR) correction was applied and ROI pairs with $Q < 0.05$ are considered as significant.

The analyses of es-TT and es-fMRI data were performed using custom MATLAB (R2019b) codes running on Windows 10 and Linux (Ubuntu 20.04) operating system in addition to the specific program listed above.

### Reporting summary

Further information on research design is available in the Nature Research Reporting Summary linked to this article.

### Data availability

es-TT data is available on Open Science Framework (OSF) database. Identifier: https://doi.org/10.17605/OSF.IO/DNGV5. es-fMRI data is available in OpenNeuro database. OpenNeuro Accession Number: ds002799 Source data are provided with this paper.

### Code availability

Custom codes used in the manuscript are available in github: https://github.com/hiroyukioya/Amygdala-Network.git.

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

## Acknowledgements

Supported by grant from National Institute of Health R01_DC004290-20 to M.A.H., National Institute of Health U01_NS103780 and the Simons Collaboration on the Global Brain to R.A. This work was conducted on an MRI instrument funded by National Institutes of Health grant 1S10OD025025-01. We thank Kirill V. Nourski, Haiming Chen, Phillip. E. Gander, Christopher Garcia, and Hiroto Kawasaki for help with experiments, John Buatti and Colin P. Derdeyn for MRI scanner logistics, Mark A. Granner for safety monitoring, and Vince A. Magnotta for MRI technical consultation.

## Author contributions

Conceptualization: H.O., M.A.H., R.A., and B.D.; methodology: H.O., M.S., B.D., and J.G.; software and formal analysis: H.O., M.S., C.K., and R.L.J.; data curation: M.S. and H.O.; writing-original draft: H.O. and M.S.; writing-review & editing: H.O., M.S., R.A., J.G., A.R., M.A.H., and B.D.; supervision: H.O. and M.A.H.; funding acquisition: R.A. and M.A.H.

## Competing interests

The authors declare no competing interests.
