## [Peer Review File · Nature Communications]

Mapping Effective Connectivity of Human Amygdala Subdivisions with Intracranial StimulationREVIEWER COMMENTS

Reviewer #1 (Remarks to the Author):

This manuscript describes results from a concurrent electrical stimulation/intracranial electroencephalography and fMRI study examining amygdala effective connectivity in neurosurgical patients. The main findings are that stimulation of the lateral amygdala produced effects in anterior cingulate, sensorimotor cortex, superior temporal sulcus and superior parietal lobule, and stimulation of the medial amygdala produced effects in orbitofrontal cortex, superior temporal gyrus, middle temporal gyrus, and prefrontal cortex. The directionality of connections was also established using these techniques. These findings were confirmed by fMRI. This is a very impressive technical accomplishment. Suggestions are provided below.

It would be helpful if the abstract included an explicit statement regarding what the novel findings (what had not been known before from animal work etc.) of the current study are.

The use of machine learning to predict amygdala stimulation site does appear to be novel. More background for this analysis and why it was undertaken could be provided in the introduction (and mentioned in the abstract).

It is a bit surprising that laterality is not specifically addressed in the main text. It would be important to include information about whether right or left amygdala were stimulated in each patient, and whether differential results were obtained from the two hemispheres. This information can only be found in the methods, and is not mentioned in the rest of the manuscript. There is an extensive literature on functional laterality of the amygdala that could be referenced and incorporated.

The text does not mention the superficial subregion of the amygdala at all. Were any recordings made at this site?

It is not clear why the conditional granger causality analyses were only conducted in one patient.

Reviewer #2 (Remarks to the Author):

Comments to Author:

This study of the human amygdala used multiple imaging and analysis approaches to probe the effective connectivity of the amygdala, allowing conclusions about the direction of responses with spatial and temporal resolution that has previously been limited in human investigations. The authors find that the cingulate cortex and prefrontal cortex are central network nodes that respond to amygdala stimulation. With the temporal resolution of their recordings, they are also able to observe differing response latencies (shorter vs. longer) between brain regions to allow inferences about causal connectivity. The conclusions of the study are strengthened by the use of multiple imaging approaches (es-TT and es-fMRI). Despite these strengths, it is not made clear the novelty of the conclusions they reach and the significance of the connectivity patterns they identify, relative to what is already known.

Major concerns:

Anatomy of location of stimulation sites:

Amygdala medial and lateral groups of sites are repeatedly referred to as the medial amygdala or lateral amygdala but these are also distinct sub nuclei of the amygdala making this terminology confusing and misleading (e.g. line 145 in results).

There appear to be medial sites that are more lateral than some that are classified as lateral (Figs 1D, 3A)?

Characteristics of patient cohort:

Alterations in network-level function of the amygdala is noted as a major hallmark of many psychiatric illnesses, but it is not indicated if any of the patients used in the study had neuropsychiatric diagnoses and if the results could have been impacted by a neuropsychiatric condition.

To address the concern of structural reorganization in long-standing epilepsy, were there consistent findings in the patients examined with shorter (2-4y) vs. longer (20+y) epilepsy?

Table 1 (page 43): aren't some of the clinical structural imaging findings for the patients in the study be major confounds? Especially ID 534 who apparently had left amygdalohippocampectomy and left temporal lobectomy (presumably an epilepsy surgical resection?).

Study design:

Sex as a biological variable is not addressed.

I was a bit concerned by the way laterality is addressed. The authors state "since we did not have any hypotheses regarding laterality, results from left and from right hemisphere are pooled here" (page 19 lines 483-484). Not having a hypothesis about laterality is not an appropriate justification for disregarding it, especially given that there is plenty of evidence for amygdala lateralization in humans, including in neuropsychiatric disorders.

The authors restricted analyses to stimulation-recordings within the same hemisphere, based on where the electrodes were placed and based on a prior report (page 19 lines 480-484). They should provide more detail about this prior report (reference 79) and why it helps justify restricting the analysis in this way.

Analysis:

How do the authors consider pre- and post-stimulation recordings within each subject to account for clinical variability between epilepsy patients (was variability in individual baselines normalized)?

Within-subjects analysis seems important to include for the 4 patients who underwent both procedures (esTT and es-fMRI) to further validate how well the results from each approach corroborates the other.

The rationale for what was considered an "abnormal" response and thus excluded from analysis (page 20 lines 506-509) was not stated and should be made clear. Additionally, this sentence was unclear: "esTT data that does not show discernible peak within the temporal windows were not used for further analysis" (page 21 lines 531-532). Is this referring to not seeing a discernible peak within a certain ROI? How would you know that there was no discernible peak due to technical errors or because the region was not responsive to stimulation for biological reasons (in which case it shouldn't be excluded from analysis)?

Minor comments:

A table of abbreviations would greatly help; in several instances terms are not defined before they are introduced. For instance: N/P, iEEG, EPs

It would be helpful to describe es-TT more generally in the introduction, and include in the methods more justification for the stimulation parameters chosen.

It would be useful to state which cited studies are NHP and thus might have a stronger relationship to what is presented in humans.

Page 21 line 536: something is missing where it currently says “---” and that sentence has some additional typos.

Reviewer #3 (Remarks to the Author):

Sawada et al. present a very valuable work on subdividing the human amygdala functionally by intracranial stimulation by means of electrophysiological as well as fMRI measures.

The authors clearly demonstrate the activation of different networks stimulating medial vs. lateral amygdala which are however, consistent networks comparing esTT and esfMRI.

Orbitofrontal and anterior cingulate cortex are shown to be key components of the amygdala network.

This MS is very timely and an important contribution not only to the amygdala community but also to brain and neuropsychological research in general.

Moreover the authors have to be congratulated in particular to

- a) contribute such an unprecedented human intra-cranial electrophysiological dataset and
- b) to backup these highly time-resolved data allowing for causality modelling by fMRI investigations which informs future fMRI results (but see caveat below).

Major points

One real potential strength of the MS is the fact that the authors utilize the high temporal resolution of EEG signals for causality analysis, which makes a very important contribution to the field. However, as we learn in the supplement, only one patient with medial stimulation is analyzed. This is a huge drawback and I do not see a reason not to validate the method / findings in more patients with 4/5/6 es-TT sites including lateral stimulation.

The MS fails short in addressing lateralization issues.

One has to expect, that an unilateral epileptic pathology does change the connectivity at a brain wide level and it will do so in a lateralized way. Therefore, I do not agree with the basic assumption of the MS, that lateralization does not play a role. Please provide evidence for that assumption!

The MS would also benefit a lot if differences regarding “structural imaging normal” and epileptic evidence patients would have been in focus and at least partially addressed. You present at several instances data of single subjects and you can pool 5 “structural imaging normal” subjects comparing to impacted ones. Can any sub-pathology characteristics be carved out? At least address this in the discussion.

Are any interactions of duration or onset of epilepsy found?

Even having in mind, that you report very valuable intracranial human data it remains unclear which patients and averages you present where in certain parts of the MS. Do we see a grand average of patients and controls? Do you mix right vs. left pathologies? Do you treat all indications identical.

It is a quite heavy job to try to follow subject numbers throughout the MS. Please at least add to Table 1 a column indicating the epilepsy patients, sorted not by ID but by the groups (structural normal, epilepsy findings) and by examination type esTT, es-fMRI.

Add type of treatment / medication.

In the same regard it is unclear at several instances showing average results what was averaged. Please add consequently this information.

In addition, it would be of great help to the reader to follow the complex manuscript if the common thread were represented better overall.

Result

Table 1: Some points already mentioned above. Moreover, please add medial / lateral stimulation sides per subject and provide at bottom average + sdev across the parameters wherever possible.

Fig. 2a and text p6,l132 to l137

Which accuracy do you report? I expect that you used balanced accuracy?

P7,l177 20 runs and 10 does not match numbers given in Fig. 3

P8,l194: I assume you mean Figure 3e

P8,l196: I assume you mean Figure 3f, g does not exist:

P8,l205: In the text you used OF, in the figure OFC.

P9,l211/212: OF and OFC used for the same

P9,l222: Why was no UMAP performed for the early N/P15 component? Please provide rationale or add it. Looks even more clear across the groups than for the P150 component.

P11,l276: please provide reference for the Leiden community algorithm.

P17,l437: Please add version of Freesurfer.

P23,l592: Coregistration to subject's T1 and MNI? These statement is unclear.

P23,l593: Do you make any use of tCompCor, CSF and white matter? From the text it appears that you calculate it but do not use it.

Discussion

Do the patients differ in their medication / treatment regime? Add that info to table 1 and comment please.

As one can see nicely in Fig. S3 the rank of latencies across the brain structure groups are highly comparable for NP15 and P150. Please discuss mechanistically how that can come along for latencies in 10th ms but also preserved in 1xx ms.

Methods

Of note, the methods part is not very detailed. It is too superficial at several points like hardware descriptions, system software's of the devices, algorithm parametrization, software versions etc.

Particular using R please provide the hardware architecture and software versions of R and the libraries used.

Were any structural non-linearities found between pre-implantation and post-implantation? Does that effect registration / (assumed) electrode position.

Cross-validation is mentioned in the main text, not in the methods.

Minor points

iEEG is not introduced before usage.

Please carefully check the usage of EEG vs iEEG (this is what you did) vs ECoG as well as partial (regressed against what) vs pearsons correlation. The terms are not to be used synonymy.

Unify abbreviations e.g. esTT or es-TT.

P3,I52: I suggest to delete EXPERIMENTAL animals.

P5,I103: Please add "in cortical regions recorded" because iEEG is dominated by cortical regions.

P21,I536: incomplete sentence

Typographical and punctuation issues throughout the MS.

Figures

Fig. 1:

B: PT 418 brain surface photography not mentioned in the legend.

Fig. 2 b,c I suggest to shift insets b and c (classifications) to the end following the descriptions in the text. Also introduce specific titles for b and c.

Fig. 2e/g ROIS are hardly visible. Please enhance e.g. lighter grey for the brain. Moreover, one thought: in red e encodes lateral > medial but red also encodes in g medial > lateral. One could use two different colors here (see e.g. usage in Fig. 3).

Fig. 3a: What does (Rt: 7, Lt: 9) encode? Not mentioned in the legends.

Fig. 3b: legend: How can a single subject single voxel time course have a. error measure? I assume we see averages over given ROI? Please clarify.

If the stimulation lasts till 30s how to explain signal reduction before stimulus ends?

Fig. 4c: Here N = number of recording channels, in Fig. 5d N =number of contacts.

Fig. 6: Which stimulation and what are the patient characteristics shown here.

Point-by-point response to the reviewers.

The reviewers' comments are in italic font with quotation marks. Responses are in bold.

NCOMMS-21-36322

("Mapping Effective Connectivity of Human Amygdala Subdivisions with Intracranial Stimulation")

Reviewer #1

General comment. *"This manuscript describes results from a concurrent electrical stimulation/intracranial electroencephalography and fMRI study examining amygdala effective connectivity in neurosurgical patients. The main findings are that stimulation of the lateral amygdala produced effects in anterior cingulate, sensorimotor cortex, superior temporal sulcus and superior parietal lobule, and stimulation of the medial amygdala produced effects in orbitofrontal cortex, superior temporal gyrus, middle temporal gyrus, and prefrontal cortex. The directionality of connections was also established using these techniques. These findings were confirmed by fMRI. This is a very impressive technical accomplishment."*

(Response) We thank the reviewer for the positive assessment of technical and scientific achievement of our study.

Comment 1. *"It would be helpful if the abstract included an explicit statement regarding what the novel findings (what had not been known before from animal work etc.) of the current study are."*

(Response) We revised the abstract and discussion (page 2 line 33-38, page 14 line 313-317) to highlight the following novel aspects of our study more clearly.

1, The key novel aspect of our study was to elucidate functional connectivity of the amygdala with a perturbational method (electrical stimulation) that permits strong causal inferences, at the whole-brain level. Prior work in humans has mapped out functional connectivity of the amygdala using resting-state fMRI, but this approach is fundamentally correlational, and limited in delineating amygdala subnuclei (due in good part to signal dropout and geometric distortion in the amygdala on BOLD-fMRI). On the other hand, prior

work in nonhuman animals has mapped out amygdala functional connectivity using causal perturbation (for instance, optogenetics in mice), but never at the whole-brain level of phasic perturbation. Our study is unique in using a perturbational, causal approach (direct intervention in the amygdala) with a whole-brain field-of-view.

2, In addition to map the spatial distribution of effective connectivity across brain, we also quantified the temporal order of these directional connections on the timescale of milliseconds (by complementing es-fMRI and es-TT). This information is important for determining the flow of information within components of the amygdala network, but has been missing in humans and again only partially applied in animals.

3, Further, we found that second-order effective connectivity between the key nodes outside of amygdala. Namely, cingulate cortex, prefrontal cortex and orbitofrontal cortex were intimately interacting in response to amygdala stimulation.

4. Finally, as the reviewer notes (see Comment 2 below), a novel finding was that we could distinguish medial from lateral amygdala by their pattern of functional connectivity, using machine-learning approaches.

Comment 2. "The use of machine learning to predict amygdala stimulation site does appear to be novel. More background for this analysis and why it was undertaken could be provided in the introduction (and mentioned in the abstract)."

(Response) Thank you for the helpful comments. As shown in the Fig. 2b and d, the es-TT response to amygdala stimulation is broadly distributed across brain and not limited to specific sites. The motivation for the machine learning analysis (decoding modeling) is to detect (potentially subtle but stable) differential information contained in this broadly distributed response.

We first applied the machine learning analysis to find what features are informative for classifying medial-vs-lateral site stimulation using EPs magnitude, latency, ROI distribution etc.

Second, we applied the similar classification analysis with unsupervised dimensional reduction in the time dimension. The evoked potential waveform has strong dependencies

along time (for example, values at time t and $t+1$ is strongly correlated). In this case, unsupervised dimensional reduction is a great way to summarize the characteristics of the waveform. Note that in this dimensionality reduction stage, no knowledge about the spatial distribution nor EPs peak latencies and amplitudes were needed.

The third application used in this manuscript is to see whether the recording brain sites (rather than the stimulation sites) could be reliably classified with the method. All three applications yielded significant classification accuracy.

We used state of the art algorithms for dimensionality reduction (UMAP) and pattern classification (CatBoost). An important point here is that both algorithms are non-linear and make few assumptions about the data; they may thus well be more sensitive to pick up relevant information in the data since temporal and spatial dependencies in the data may well be non-linear. Further, CatBoost algorithm support categorical input feature (such as ROI labels, which were found to be very important for the classification).

These points have been added in the introduction and mentioned in the abstract (page 2 line 24-25 and page 5 line 91-92). We put some of the details of this method in Methods section (page 23 line 583-588, page 24 line 601-602 and line 605-610).

Comment 3. "It is a bit surprising that laterality is not specifically addressed in the main text. It would be important to include information about whether right or left amygdala were stimulated in each patient, and whether differential results were obtained from the two hemispheres. This information can only be found in the methods, and is not mentioned in the rest of the manuscript. There is an extensive literature on functional laterality of the amygdala that could be referenced and incorporated."

(Response) We agree that this is an important point, also raised in some of the other reviews. In our revised manuscript, we now cite some of the background that the reviewer alludes to.

While behavioral studies in animals using pharmacological manipulation of the amygdala have suggested there may be functional lateralization of amygdala (Alvarez and Banzan, 2011; Coleman-Mesches and McGaugh, 1995; Sullivan et al., 2009), these studies do not yet provide a consistent picture that fits with functional imaging studies in humans. The most relevant study is the one by Grayson et al (Grayson et al., 2016) in which pharmacological inactivation of the amygdala in monkeys was combined with whole-brain fMRI. In that study,

perturbing the left or the right amygdala produced extremely similar patterns of functional connectivity changes between the amygdala and the rest of the brain (see Fig. 3 in that paper). In humans, functional MRI studies show varying degree of lateralization of activation in the amygdala to stimuli that carry emotional and social information (Freeman et al., 2014; Hardee et al., 2008; Wang et al., 2017), for review (Baas et al., 2004; Sergerie et al., 2008). However, those studies did not investigate what our study investigates, which is functional connectivity rather than simply magnitude of amygdala activation. A study using a relatively large subject sample size (Bickart et al., 2012) looked at amygdala resting-state functional connectivity and did not find substantial left-right differences; neither did a recent study that had arguably the best data quality of any of these (Sylvester et al., 2020). No prior study in human has systematically investigated laterality of amygdala connectivity and what qualitative difference between hemisphere reported remain unclear (Bzdok et al., 2013; Mishra et al., 2014).

Our conclusion from the extant literature is that (a) to the extent that there are left-right differences in amygdala activation (and the literature is inconsistent here and often based on underpowered studies), these exist for the magnitude of activation in task-fMRI studies, which is a rather different metric than the focus of our study (and depends critically on the use of statistical thresholds in the analysis). By contrast (b) resting-state functional connectivity studies of the amygdala (more similar to our metric) do not generally report laterality effects. Our study is a functional connectivity study (and psychologically closest to resting-state, since our electrical stimulation did not evoke any cognitive effects). We consequently do not feel that the literature would provide justification for any hypothesis about laterality, and we consequently tested none.

Finally, another question also pointed out by another reviewer is that sex and hemispheric lateralization may interact (Cahill, 2006). This interaction has been reported in humans and animals (Gupta et al., 2011; Kilpatrick et al., 2006; Sullivan et al., 2009; Tranel and Bechara, 2009). This interaction is again claimed only for magnitude of amygdala activation, not functional connectivity.

We appreciate these questions, and we fully agree that in principle laterality (and subject sex) are important variables to consider. However, in the absence of a clear hypothesis, and with our very small sample sizes and other sources of variability, in our view, it would not be appropriate to conduct null-hypothesis significance testing with respect to these variables. Nonetheless, we agree that it would be informative simply to explore these questions in our data and report the results. Consequently, we now report effect size

analyses of sex and laterality. We report only effect sizes in the paper and refrain from reporting p-values since we feel this would be misleading for the reason noted above. Results from the analyses below are meant to be a guide for the reviewer's interpretation of the effects presented in the paper. Larger samples (number of patients) in future studies will be required to build on these exploratory findings with sufficient statistical reliability.

We have therefore performed additional analyses with hemisphere and sex as the key variables (Male patients N = 6, Female patients N = 7. For Male, 11 right and 11 left and for Female, 22 right and 7 left es-TT sessions). We show the effect size of sex and hemisphere, and presented below as a summary table and a figure. This figure is presented in the paper as Supplementary Fig. S8.

The P150 response was generally larger in males than females as can be seen in the bottom row in the figure. But there is no clear tendency for hemispheric lateralization in either N/P15 or P150. The data also suggest potential presence of a Sex – Hemisphere interaction especially in the early component (N/P15). Although this potential interaction seems to be present in widespread brain areas, the effect was most pronounced in OF, CC, TL and SM ROI groups.

This point is added in the manuscript (in results section page 11 line 243 – 255 , in discussion page 16 line 375 - 381, in methods section page 25 line 620 - 623).

Information of side of laterality of stimulation for each patient is added in Supplementary Table S1.

Figure S8. Effect of Sex and Hemisphere. es-TT response magnitudes (mean+SE) showing effect sizes of two factors (Sex and Hemisphere) for each ROI group. X-axis: Sex, Y-axis: mean es-TT EPs response amplitude in sd unit. N = Number of channels. Medial and lateral group stimulation are combined.

Summary table (mean and SE are reported)

	Hemi	Gender	ROI group						Mean
			OF	IPFC	CC	SM	TL	PL	
N/P15	Rt	Male	14.7 (1.4)	5.2 (0.3)	6.7 (1.7)	5.3 (0.8)	14.6 (1.1)	10.6 (1.7)	9.5 (1.2)
		Female	6.7 (0.9)	3.1 (0.1)	2.9 (0.2)	2.7 (0.3)	5.6 (0.2)	2.9 (0.1)	4 (0.3)
	Lt	Male	8.3 (1.6)	5.0 (0.4)	7.3 (1.8)	4.9 (0.5)	10.3 (0.7)	5.1 (0.4)	6.8 (0.9)
		Female	16.0 (2.7)	5.7 (0.4)	7.6 (1.6)	4.5 (0.5)	8.4 (0.9)	3.6 (0.3)	7.6 (1.1)
		Mean	11.4 (1.7)	4.8 (0.3)	6.1 (1.3)	4.4 (0.5)	9.7 (0.7)	5.6 (0.6)	7.0 (0.9)
P150	Rt	Male	27.9 (1.7)	29.3 (1.4)	56.7 (5.4)	23.4 (4.9)	24.9 (1.5)	20.4 (3.7)	30.4 (3.1)
		Female	12.5 (1.6)	12.6 (1.1)	20.6 (1.9)	12.5 (0.8)	14.8 (0.5)	11.3 (0.8)	14.1 (1.1)
	Lt	Male	19.0 (3.5)	35.0 (2.3)	20.5 (2.8)	17.4 (1.6)	26.1 (1.5)	21.3 (1.9)	23.2 (2.3)
		Female	13.8 (1.9)	9.7 (0.8)	15.6 (4.3)	7.2 (0.9)	11.0 (0.9)	7.1 (0.6)	10.7 (1.6)
		Mean	18.3 (2.2)	21.7 (1.4)	28.4 (3.6)	15.1 (2.1)	19.2 (1.1)	15.0 (1.8)	19.6 (2.0)

Comment 4. *“The text does not mention the superficial subregion of the amygdala at all. Were any recordings made at this site?”*

(Response) Since we used clinical macro-electrodes for stimulation within the amygdala, we did not attempt to examine *intra-amygdala* connectivity by recording from those same contacts. In some cases, there were additional contacts that were not used for stimulation in the medial part of the amygdala. But due to the very close proximity to the stimulation site, recording from those contacts had large artifacts and not suitable for formal analysis. We think, in humans, the study for this micro-circuit should be done using micro-stimulation with high-density micro-electrodes. We mentioned this point in the revised text (page 17, line 400 – 405).

We used amygdala parcellation from CIT168 atlas (see method). The subnuclei called as central and medial nuclei (CMN) in our paper corresponds to superficial group. We now also mentioned this point in the revised text (page 6, line 114).

Comment 5. *“It is not clear why the conditional granger causality analyses were only conducted in one patient.”*

(Response) We conducted this analysis in only one patient, because this was the only patient who had suitable data for the Granger causality analysis. Specifically, this analysis requires that intracranial electrode should cover the ROI for the key nodes at single subject level. This patient had electrode coverage within all the brain parcellation of our interest (anterior cingulate cortex, prefrontal cortex, sensory-motor cortex, parietal and lateral temporal lobe).

A second reason was that the patient had multiple data sets (experiments) available, giving us substantially more samples. Importantly, Conditional Granger causality (CGC) analysis does *not* allow trial rejection for *some* of the channels in the dataset per trials because the analysis uses data from all channels (in other words, it is *not* a pair-wise connectivity measure) . For example, we cannot perform the analysis on any trials whose data was rejected in any of the channels. So for the CGC analysis including many channels, it requires fairly large number of trials.

We have further looked into the patients in whom multiple experiments were available with electrode coverage in the ROIs of interest. We found that PT384 (lateral and medial

stimulation) and PT515 (only lateral stimulation) datasets were potentially suited to the analysis and ran the same Conditional Granger Causality analyses and presented. The additional results are now presented in Fig. 6 and Supplementary Fig. S7.

Of note, the electrode placement was different in each patient for clinical reason. So even in the same ROI group, the actual recording location differs across patients. The number of es-TT runs also varied. These factors (including difference in trial numbers as mentioned above) as well as individual difference could potentially influence the CGC results. However, the new results in fact confirmed the findings presented in the initial submission. ACC, OFC and PFC are heavily connected bidirectionally (but asymmetrically) during medial amygdala stimulation. ACC - temporal lobe connectivity was again detected as in the initial manuscript.

Compared to the medial amygdala group stimulation, Conditional Granger Causality applied to the lateral group stimulation showed marked differences in ACC – PFC (dorsolateral part) connectivity (greatly reduced). On the other hand, ACC – OFC connectivity remained as prominent as with medial group stimulation.

We revised texts in the corresponding part of methods (page 26 , line 697 - 699), results (page 12 line 269 - 270, line 276 - 279 and line 283 - 286) and discussion sections (page 14 line 329 - 330).

Reviewer #2

General comment. *“This study of the human amygdala used multiple imaging and analysis approaches to probe the effective connectivity of the amygdala, allowing conclusions about the direction of responses with spatial and temporal resolution that has previously been limited in human investigations. The authors find that the cingulate cortex and prefrontal cortex are central network nodes that respond to amygdala stimulation. With the temporal resolution of their recordings, they are also able to observe differing response latencies (shorter vs. longer) between brain regions to allow inferences about causal connectivity. The conclusions of the study are strengthened by the use of multiple imaging approaches (es-TT and es-fMRI). Despite these strengths, it is not made clear the novelty of the conclusions they reach and the significance of the connectivity patterns they identify, relative to what is already known.”*

(Response) Thank you for your comments on the manuscript. We agree that the novelty presented in the manuscript could be clearer. We revised the abstract and discussion (page 2 line 33 - 38, page 14 line 313 - 317) to highlight the following novel aspects of our study more clearly.

1, The key novel aspect of our study was to elucidate functional connectivity of the amygdala with a perturbational method (electrical stimulation) that permits strong causal inferences, at the whole-brain level. Prior work in humans has mapped out functional connectivity of the amygdala using resting-state fMRI, but this approach is fundamentally correlational, and limited in delineating amygdala subnuclei (due in good part to signal dropout and geometric distortion in the amygdala on BOLD-fMRI). On the other hand, prior work in nonhuman animals has mapped out amygdala functional connectivity using causal perturbation (for instance, optogenetics in mice), but never at the whole-brain level of phasic perturbation. Our study is unique in using a perturbational, causal approach (direct intervention in the amygdala) with a whole-brain field-of-view.

2, In addition to map the spatial distribution of effective connectivity across brain, we also quantified the temporal order of these directional connections on the timescale of milliseconds (by complementing es-fMRI and es-TT). This information is important for determining the flow of information within components of the amygdala network, but has been missing in humans and again only partially applied in animals.

3, Further, we found that second-order effective connectivity between the key nodes outside of amygdala. Namely, cingulate cortex, prefrontal cortex and orbitofrontal cortex were intimately interacting in response to amygdala stimulation.

4. Finally, as one of the reviewer notes (see Comment 2 of reviewer #1), a novel finding was that we could distinguish medial from lateral amygdala by their pattern of functional connectivity, using machine-learning approaches.

Comment 1. *“Anatomy of location of stimulation sites:*

Amygdala medial and lateral groups of sites are repeatedly referred to as the medial amygdala or lateral amygdala but these are also distinct sub nuclei of the amygdala making this terminology confusing and misleading (e.g. line 145 in results). There appear to be medial sites that are more lateral than some that are classified as lateral (Figs 1D, 3A)?”

(Response) We apologize for any confusion in the terminology we used in the manuscript. Indeed, as the reviewer correctly notes, we are not resolving specific nuclei, but rather group of nuclei (more macroscopic regions within the amygdala). We made changes in the manuscript to avoid the usage of “medial amygdala” and “lateral amygdala”. Instead, we now use “medial group” and “lateral group” for indicating amygdala stimulation sites. We now mentioned this point in introduction (page 4 line 56 - 60).

The reviewer’s point that some medial and lateral sites may be misclassified may be due to different inter-contact distances and electrode trajectory. Fig. 1d and 3a show midpoint of two adjacent stimulated contacts and we confirmed that the stimulation site grouping was correct as stated in the manuscript.

Comment 2. “Alterations in network-level function of the amygdala is noted as a major hallmark of many psychiatric illnesses, but it is not indicated if any of the patients used in the study had neuropsychiatric diagnoses and if the results could have been impacted by a neuropsychiatric condition.”

(Response) This is a great point. We now closely assessed the psychiatric status of our subjects. While none had major diseases such as schizophrenia, autism, or Alzheimer’s disease, we did find varying levels of two major neuropsychiatric condition known to have involvement of amygdala, that is, depression and anxiety disorders. Both conditions have common clinical features such as in onset time (adolescence) and gender preference (greater risk for female than male). Co-occurrence of the two conditions was reported to be significantly high, and anxiety symptoms have been suggested as a significant predictor of occurrence of depression (Angold et al., 1999; Kravitz et al., 2014). Given this relationship of clinical characteristics between the two, we pooled the two conditions and compared the data from patient who had either of these conditions and the data from the patient without these conditions. We used neuropsychological assessment performed by clinical neuropsychologists to quantify these before electrode implantation surgery. This information is added to the new column in Table S1.

Next, we examined whether es-TT response differ significantly by the presence or absence (N = 5 and 8, respectively) of either depression or anxiety. Statistical tests were performed per each ROIs (ttest, Bonferroni correction) between depression/anxiety positive and negative subgroups and results are now presented in Supplementary Fig. S9. We found significant differences in dorsomedial and dorsolateral prefrontal cortices (dmPFC and dIPFC including superior and middle frontal gyrus), superior and middle temporal gyri and

angular gyrus ($P < 0.05$, ttest, Bonferroni corrected). Importantly, prefrontal cortex showed significantly larger es-TT response in the depression/anxiety (+) group whereas the middle and superior temporal gyri showed significantly smaller response in that group. This point is now discussed in the paper (page 16 line 382 - 385).

We also checked the latency distribution across ROI groups as we presented in the main manuscript (Fig. 4c and 5d) focusing on the presence of depression/anxiety. These results are from combined data (medial and lateral group stimulation). As shown in the bargraphs (mean+se are shown) below, the general tendency (CC, IPFC and OF show short latency) was quite well preserved in this subgroup. Note that there were only 2 contacts in CC group (cingulate gyri) for this subgroup for P150. There were significant differences between the two conditions especially in P150 in TL group where there was also a magnitude difference. This shows that although response magnitude showed interesting differences between depression/anxiety (+) and (-) groups, the order of timing of response propagation was quite similar to the overall latency results presented in the main paper.

While we found the above interesting effects of depression/anxiety, we are careful to draw generalizable conclusions at this stage, given that all patients had epilepsy to begin with, and given our relatively small sample size. This point is also mentioned in the discussion (page 16 line 382 – 385, as mentioned above) and in results (page 11 line 256 - 265).

Depression & Anxiety

Figure S9. Amygdala stimulation es-TT response in patients with depression and/or anxiety. (a) ROIs that showed significant difference between patients with and without either depression or anxiety disorder (N = 5 and 8, respectively) is color-coded and shown. ROIs in white mesh indicated non-significant ROIs. **(b)** es-TT response averaged over all ROIs for N/P15 (left) and P150 (right). Mean and SE are shown. Number of valid contacts are also shown. **(c)** Same as (b), but ROIs that showed significant difference ($P < 0.05$, Bonferroni correction). Medial and lateral group stimulations were combined for the analyses.

Peak latencies averaged within each ROI group are shown grouped for positive depression or anxiety (Depression/Anxiety(+)) and negative depression or anxiety (Depression/Anxiety(-)). Medial and lateral group stimulations are combined. Mean+SE are shown. Asterisk indicate significant difference between the two groups.

Comment 3. “To address the concern of structural reorganization in long-standing epilepsy, were there consistent findings in the patients examined with shorter (2-4y) vs. longer (20+y) epilepsy?”

(Response) This is a good question. We performed additional analysis to see if there is any significant impact of duration of epilepsy on the amygdala’s connectivity contrasting (< 4 yrs.) vs (>20 yrs.) as suggested.

Results are presented as Supplementary Fig. S10. Statistical tests (ttests with Bonferroni correction) indicated that there was a significant differences between the two groups (but much more restricted than the full set of results presented in the main text in the initial submission), specifically in middle temporal gyrus (p150). Given the small sample size within the subgroups and the vastly varying clinical presentations at individual level, much larger cohorts of subjects will be required to answer this question reliably.

We additionally examined whether the response patterns differed among various sub-grouping of patients (by the duration of epilepsy or structural abnormality in the MRI). The results are presented below (Supplementary Fig. S12). This scatterplot matrix shows the es-TT responses are well correlated across various sub-groupings, suggesting relatively stable results irrespective of epilepsy duration or structural brain abnormality.

The issue on the sub-groupings is discussed in discussion section (page 16 line 386 - 393).

Supplementary Figure S10. ROI averaged intracranial EPs waveform grouped by duration of epilepsy. (a) EPs magnitude distributions mapped on the MNI brain for N/P15 and P150 for epilepsy duration over 20 years and <4 years group. **(b)** Group averaged es-TT response mapped on to MNI brain. **(b)** ROI EPs that showed significant difference between groups. Red line and area: Overall averaged waveform and its se (Epilepsy duration < 4 yrs.). Black line and area: Overall averaged waveform and its se (Epilepsy duration > 20 yrs.). Red dots on the waveform indicate significantly different N/P15 (dots at 15 ms) or P150 component (dots at 150 ms) of EPs between two conditions ($P < 0.05$, Bonferroni correction). Only shows ROIs with number of contacts > 20 for each condition. Number of valid contacts within the ROI are also shown. Only ROI that showed significant difference is shown. Medial and lateral group stimulation were combined for the analyses.

Correlations of es-TT response among sub-groups. Each dot represents response magnitude (in sd unit) averaged within a ROI. Pearson correlation values were calculated and indicated by colored box.

Supplementary Figure S12. es-TT response magnitude distribution averaged within each structural ROIs and comparisons across different sub-groupings. Pearson correlation values (r) larger than 0.25 were indicated by colored axis-box.

Comment 4. "Table 1 (page 43): aren't some of the clinical structural imaging findings for the patients in the study be major confounds? Especially ID 534 who apparently had left amygdalohippocampectomy and left temporal lobectomy (presumably an epilepsy surgical resection?)."

(Response) This is also a great question. We carried out additional analyses comparing evoked potential (EP) magnitudes between a structural imaging normal group ($N = 3$ for es-TT) and a structural lesion group ($N = 10$), results from which are now presented in Supplementary Fig. S11. There were significant differences between the two groups for N/P15 in precentral gyrus and for P150 in middle frontal gyrus. Given limited number of data points in the structural normal group, further subdivision in hemisphere could not be done. There was a tendency for having a clear EP peak in the structural normal group.

While we have quantified the patient groups suggested by the reviewer's comment, we note that larger sample sizes would be required to draw any conclusions, for two reasons. First,

there is extreme variability in structural imaging findings, as can be seen in the table S1 and typical for our patient population. Second, there is a very small number of patients in the structural normal group (also consistent with our patient population). On the other hand, given large variability of imaging findings, individually specific contributions will be substantially averaged out – the results we present are thus ones that are robust to these individual differences. The issue on the sub-groupings is discussed in discussion section (page 16 line 386 - 393).

Yes, PT534 had prior surgery on the left for the treatment of epilepsy. In this patient, right (intact) amygdala was stimulated but most of the coverage was on the left side (ipsilateral contact N = 28, contra-lateral contacts N > 200). Since we analyzed only ipsilateral connectivity, we believe that inclusion of this patient did not introduce any bias.

Supplementary Figure S11. ROI averaged intracranial EPs waveform grouped by structural MRI findings. (a) EPs magnitude distributions mapped on the MNI brain for N/P15 and P150 for structural normal and abnormal group. (b) Group averaged es-TT response mapped on to MNI brain. Red line and area: Overall averaged waveform and its se (Structural MRI normal). Black line and area: Overall averaged waveform and its se (Structural MRI abnormal). Red dots on the

waveform indicate significantly different N/P15 (dots at 15 ms) or P150 component (dots at 150 ms) of EPs between two conditions ($P < 0.05$, Bonferroni correction). Only shows ROIs with number of contacts > 20 for each condition. Number of valid contacts within the ROI are also shown. Only ROIs that showed significant difference are shown. Medial and lateral group stimulations were combined for the analyses.

Comment 5. "Sex as a biological variable is not addressed."

(Response) We thank the reviewer for the opportunity to address this important point. We have added new analyzes of our data with sex as a factor. Since this point is closely related to the reviewer's comment 6 below, we combined our response to comments 5 and 6 (please see our response to comment 6).

Comment 6. "I was a bit concerned by the way laterality is addressed. The authors state "since we did not have any hypotheses regarding laterality, results from left and from right hemisphere are pooled here" (page 19 lines 483-484). Not having a hypothesis about laterality is not an appropriate justification for disregarding it, especially given that there is plenty of evidence for amygdala lateralization in humans, including in neuropsychiatric disorders."

(Response) We agree that this is an important point, also raised in some of the other reviews.

While behavioral studies in animals using pharmacological manipulation of the amygdala have suggested there may be functional lateralization of amygdala (Alvarez and Banzan, 2011; Coleman-Mesches and McGaugh, 1995; Sullivan et al., 2009) , these studies do not yet provide a consistent picture that fits with functional imaging studies in humans. The most relevant study is the one by Grayson et al (Grayson et al., 2016) in which pharmacological inactivation of the amygdala in monkeys was combined with whole-brain fMRI. In that study, perturbing the left or the right amygdala produced extremely similar patterns of functional connectivity changes between the amygdala and the rest of the brain (see Fig. 3 in that paper). In humans, functional MRI studies show varying degree of lateralization of activation in the amygdala to stimuli that carry emotional and social information (Freeman et al., 2014; Hardee et al., 2008; Wang et al., 2017), for review (Baas et al., 2004; Sergerie et al., 2008). However, those studies did not investigate what our study investigates, which is functional connectivity rather than simply magnitude of amygdala activation. A study using

a relatively large subject sample size (Bickart et al., 2012) looked at amygdala resting-state functional connectivity and did not find substantial left-right differences; neither did a recent study that had arguably the best data quality of any of these (Sylvester et al., 2020). No prior study in human has systematically investigated laterality of amygdala connectivity and what qualitative difference between hemisphere reported remain unclear (Bzdok et al., 2013; Mishra et al., 2014).

Our conclusion from the extant literature is that (a) to the extent that there are left-right differences in amygdala activation (and the literature is inconsistent here and often based on underpowered studies), these exist for the magnitude of activation in task-fMRI studies, which is a rather different metric than the focus of our study (and depends critically on the use of statistical thresholds in the analysis). By contrast (b) resting-state functional connectivity studies of the amygdala (more similar to our metric) do not generally report laterality effects. Our study is a functional connectivity study (and psychologically closest to resting-state, since our electrical stimulation did not evoke any cognitive effects). We consequently do not feel that the literature would provide justification for any hypothesis about laterality, and we consequently tested none.

Finally, another question also pointed out by another reviewer is that sex and hemispheric lateralization may interact (Cahill, 2006). This interaction has been reported in humans and animals (Gupta et al., 2011; Kilpatrick et al., 2006; Sullivan et al., 2009; Tranel and Bechara, 2009). This interaction is again claimed only for magnitude of amygdala activation, not functional connectivity.

We appreciate these questions, and we fully agree that in principle laterality (and subject sex) are important variables to consider. However, in the absence of a clear hypothesis, and with our very small sample sizes and other sources of variability, in our view, it would not be appropriate to conduct null-hypothesis significance testing with respect to these variables. Nonetheless, we agree that it would be informative simply to explore these questions in our data and report the results. Consequently, we now report effect size analyses of sex and laterality. We report only effect sizes in the paper and refrain from reporting p-values since we feel this would be misleading for the reason noted above. Results from the analyses below are meant to be a guide for the reviewer's interpretation of the effects presented in the paper. Larger samples (number of patients) in future studies will be required to build on these exploratory findings with sufficient statistical reliability.

We have therefore performed additional analyses with hemisphere and sex as the key variables (Male patients N = 6, Female patients N = 7. For Male, 11 right and 11 left and for Female, 22 right and 7 left es-TT sessions). We show the effect size of sex and hemisphere, and presented below as a summary table and a figure. This figure is presented in the paper as Supplementary Fig. S8.

The P150 response was generally larger in males than females as can be seen in the bottom row in the figure. But there is no clear tendency for hemispheric lateralization in either N/P15 or P150. The data also suggest potential presence of a Sex – Hemisphere interaction especially in the early component (N/P15). Although this potential interaction seems to be present in widespread brain areas, the effect was most pronounced in OF, CC, TL and SM ROI groups.

This point is added in the manuscript (in results section page 11 line 243 – 255, in discussion page 16 line 375 - 381, in methods section page 25 line 620 - 623).

Information of side of laterality of stimulation for each patient is also added in Table S1.

Figure S8. Effect of Sex and Hemisphere. es-TT response magnitudes (mean+SE) showing effect sizes of two factors (Sex and Hemisphere) for each ROI group. X-axis: Sex, Y-axis: mean es-TT EPs response

amplitude in sd unit. N = Number of channels. Medial and lateral group stimulation are combined. Medial and lateral group stimulation were combined for the analyses.

Summary table (mean and SE are reported)

	Hemi	Gender	ROI group						Mean
			OF	IPFC	CC	SM	TL	PL	
N/P15	Rt	Male	14.7 (1.4)	5.2 (0.3)	6.7 (1.7)	5.3 (0.8)	14.6 (1.1)	10.6 (1.7)	9.5 (1.2)
		Female	6.7 (0.9)	3.1 (0.1)	2.9 (0.2)	2.7 (0.3)	5.6 (0.2)	2.9 (0.1)	4 (0.3)
	Lt	Male	8.3 (1.6)	5.0 (0.4)	7.3 (1.8)	4.9 (0.5)	10.3 (0.7)	5.1 (0.4)	6.8 (0.9)
		Female	16.0 (2.7)	5.7 (0.4)	7.6 (1.6)	4.5 (0.5)	8.4 (0.9)	3.6 (0.3)	7.6 (1.1)
		Mean	11.4 (1.7)	4.8 (0.3)	6.1 (1.3)	4.4 (0.5)	9.7 (0.7)	5.6 (0.6)	7.0 (0.9)
P150	Rt	Male	27.9 (1.7)	29.3 (1.4)	56.7 (5.4)	23.4 (4.9)	24.9 (1.5)	20.4 (3.7)	30.4 (3.1)
		Female	12.5 (1.6)	12.6 (1.1)	20.6 (1.9)	12.5 (0.8)	14.8 (0.5)	11.3 (0.8)	14.1 (1.1)
	Lt	Male	19.0 (3.5)	35.0 (2.3)	20.5 (2.8)	17.4 (1.6)	26.1 (1.5)	21.3 (1.9)	23.2 (2.3)
		Female	13.8 (1.9)	9.7 (0.8)	15.6 (4.3)	7.2 (0.9)	11.0 (0.9)	7.1 (0.6)	10.7 (1.6)
		Mean	18.3 (2.2)	21.7 (1.4)	28.4 (3.6)	15.1 (2.1)	19.2 (1.1)	15.0 (1.8)	19.6 (2.0)

Comment 7. “The authors restricted analyses to stimulation-recordings within the same hemisphere, based on where the electrodes were placed and based on a prior report (page 19 lines 480-484). They should provide more detail about this prior report (reference 79) and why it helps justify restricting the analysis in this way.”

(Response) This is a great point related to the hemispheric lateralization issue.

A first reason we restricted analyses within the ipsilateral connection in the manuscript was that the direct interhemispheric connection from the amygdala is thought to be considerably weaker (if existing at all) than connections confined to the ipsilateral hemisphere. The paper (Wilson et al., 1991) describes results of electrical stimulation evoked response in humans. An important finding of this report is that the amygdala stimulation induced responses exclusively in the ipsilateral (not in the contralateral) limbic structures. Another paper (Demeter et al., 1990) using tracer injection in non-human primate shows direct interhemispheric projection from the amygdala were very sparse. Given these prior reports, direct interhemispheric connection from amygdala seems to be a minor component.

A second reason is that our analyses are particularly interested in examining the precise timing of es-TT responses on the millisecond scale. This is a crucial and unique piece of information of our dataset. Inclusion of potential inter-hemispheric connection (responses contra-lateral to the stimulated hemisphere) would complicate the timing analysis and its interpretation (contralateral cortical responses could arise from direct effects of the stimulated amygdala on the contralateral amygdala, but could arise from cortico-cortical effects through the corpus callosum or anterior commissure at multiple levels). These points are now included in text (method section, page 21 line 519 - 526).

Comment 8. “How do the authors consider pre- and post-stimulation recordings within each subject to account for clinical variability between epilepsy patients (was variability in individual baselines normalized)?”

(Response) Averaging over many trials significantly reduced non-phase locked signal both in pre- and post-stimulation recordings. Since EPs are a phase locked component of the signal, this significantly increase SNR and reduce the variability of the baseline period. Additionally, we normalized the EPs waveform using their standard deviation within the baseline period (pre-stimulation period) for every channel separately. These procedures reduced the inter-individual as well as inter-contact variability. This point is described in Method section (page 22 line 552 - 553).

Comment 9. “Within-subjects analysis seems important to include for the 4 patients who underwent both /procedures (esTT and es-fMRI) to further validate how well the results from each approach corroborates the other.”

(Response) We agree that es-TT and es-fMRI data obtained from the same subjects can be very useful and thank the reviewer for the suggestion. We have performed the additional analyses for answering this point. Data from es-TT and es-fMRI from the same subjects were analyzed with regression analysis. As we responded to the reviewer’s comment 7 above, we focused on the patient who had ipsilateral response to the stimulated hemisphere in both es-TT and es-fMRI experiments (N = 2). As seen in Supplementary Fig. S6, the amygdala stimulation induced electrophysiological and BOLD response obtained from the same subject showed significant correlations both in early and late time windows. An important future direction would be to examine this relationship in the regions we found

a negative BOLD response in es-fMRI. These regions lacked electrode coverage in our case, making comparisons between es-fMRI and es-TT impossible.

Comment 10. “The rationale for what was considered an “abnormal” response and thus excluded from analysis (page 20 lines 506-509) was not stated and should be made clear.”

(Response) The general rationale for exclusions were to reject noisy trials that contain inter-ictal epileptic spikes and non-physiological artifacts such as cable motion artifact or long decay artifact from amplifier saturation. These potentials usually have distinct large amplitude, and the amplitude thresholding procedure used is commonly adopted to clean the electrophysiological recordings.

The term “*abnormal*” used in the initial manuscript was confusing since that may imply an abnormal neural response due to underlying pathology or dysfunctional brain (rather than the artifacts mentioned above). We have made changes in the text and explained this point more clearly (page 22 line 546 - 548). Note that we also discarded electrodes that were placed in the site that contributed to the seizure generation across all our analyses, thus eliminating recordings obtained from pathological tissue.

An image that shows samples of rejected single trial waveforms is attached below. These trials were contaminated with non-physiological noise and rejected according to the criterion mentioned in the text. Electrical stimulation was delivered at time 0.

Comment 11. “this sentence was unclear: “esTT data that does not show discernible peak within the temporal windows were not used for further analysis” (page 21 lines 531-532). Is this referring to not

seeing a discernible peak within a certain ROI? How would you know that there was no discernible peak due to technical errors or because the region was not responsive to stimulation for biological reasons (in which case it shouldn't be excluded from analysis)?”

(Response) We apologize that this was unclear. We meant to indicate cases where there was no response at *contact level*.

The evoked potentials have typical characteristic of waveform deflection starting right after the stimulus. We examined the evoked potential's spatial distribution across brain. If we see evoked potentials in response to stimulus for some channels but not others, we consider the recording setup itself was working fine. We also checked whether we could observe stimulus artifacts in the recording (channel by channel) to check whether electrical stimulation was actually delivered without cabling or connection problems.

Another common technical problem was related to electrodes. Our recording system (Neuralynx ATLAS) splits iEEG signal for both research and clinical use allowing on-line monitoring of all recording channels both at a research workstation and a clinical epilepsy monitoring system. So the obvious malfunction of stimulator or lead breakage could be picked up on-line during the experiment. In the case of unstable or disconnected electrode leads of either reference or recording contacts, this could also be easily picked up off-line by examining single trial waveforms for existence of high-frequency noise contamination, large baseline fluctuations or flat lines. And the EPs does not show clear peak in the time windows of interest. Examining single trials together with EPs (averaged potentials) usually gives us clear idea about the recordings.

Our data were averaged field potentials recorded from low-impedance electrodes and this gives us better SNR compared to high-impedance unit recording. In the case of iEEG recordings from high-impedance contacts for multi- or single- unit analysis, we agree that it is sometimes difficult to distinguish the cause for not observing apparent response; whether the tissue is not responding or recording has some technical issue (lead breakage, electrodes were in white matter, connection problem or very noisy recordings etc.). However, due to the reason we discussed above, technical problem could be relatively easily picked up in our recordings.

EPs (= averaged field potential) response is characterized by its amplitude and latency of significant deflection from the baseline. If there is no peak because of any biological reasons (e.g., no functional connectivity), it is impossible to reliably extract responses in

terms of its amplitude and latency (no discernible peak means no latency). Since we are especially concerned about the latency of the responses, we've intended to use only channels in which we could unambiguously determine their latencies. Consequently, we did not include the data from those contacts that did not show clear responses even there is no technical problem. We understand that this might ignore sub-threshold (non-significant) values in the analyses but we believe that inclusion of unreliable values obtained from the contacts that did not show supra-threshold response (indistinguishable from background noise and suggests no connectivity) into the following analysis pipeline might increase the noise in the results.

This point is explained in method section (page 23 line 571-573).

Comment 12. "A table of abbreviations would greatly help; in several instances terms are not defined before they are introduced. For instance: N/P, iEEG, EPs"

(Response) Thank you for this suggestion and sorry for missing abbreviations. We added the table for abbreviations used in the manuscript.

Comment 13. "It would be helpful to describe es-TT more generally in the introduction, and include in the methods more justification for the stimulation parameters chosen."

(Response) We now clarify in the introduction that es-TT is a method that probes directional connectivity in the brain and differs from more prolonged electrical stimulation to study focal function of the stimulated site by inducing behavioral change (as is typically done for clinical reasons). We also modified sentences in the abstract (page5 line 71 - 75).

Details of the stimulus parameter selection are now provided in the methods section (page 20 line 499 - 500, page 21 line 504 - 511). Namely, these parameters of electrical stimulation were determined by the following factors: 1, Safety. 2, Efficacy of neural stimulation. 3, Low artifact contamination from stimulation.

1, There are empirical safety limits based on the charge density and charge/phase usually represented as Shannon's plot which is attached below. Our stimulation parameters are indicated by the black square in the plot below and fall into the safe region (non-highlighted area). The balance of injected charge is important, and we used a charge-balanced waveform to avoid unintended local charge accumulation. Also, the repetition rate used in

our es-TT (single pulse repeated every 2 second) is very slow and accumulation of the charge should not be a problem.

2, The efficacy of the stimulation has been demonstrated by the papers published from other labs (referenced in the manuscript). Our stimulation parameters (9 mA, 0.2 ms/phase) were in the range reported (3 -10 mA amplitude, 0.1 - 0.5 ms/phase). A study in human motor cortex using direct cortical stimulation found that Chronaxie (2 times the Rheobase which indicates asymptotic current threshold for excitation) was around 0.2 ms which is exactly what we used (Abalkhail et.al., Clinical Neurophysiology, 2017) and this corresponds well with the value found in nonhuman primate studies as well (0.18 ms in V1, Tehovnik, et.al., J, Neurophys, 2006).

3, This brief stimulus also enabled us to analyze responses with fast latency.

Comment 14. "It would be useful to state which cited studies are NHP and thus might have a stronger relationship to what is presented in humans."

(Response) We have revised the text so that this point is clear throughout the manuscript (page 4 line 51 - 52, line 63, line 65 – 66, and page21 line 508)

Comment 15. "page 21 line 536: something is missing where it currently says "---" and that sentence has some additional typos."

(Response) We are sorry about this mistake. We have corrected this sentence (now page 23, line 577 - 578).

Reviewer #3

General comment. *“This MS is very timely and an important contribution not only to the amygdala community but also to brain and neuropsychological research in general.*

Moreover the authors have to be congratulated in particular to

a) contribute such an unprecedented human intra-cranial electrophysiological dataset and

b) to backup these highly time-resolved data allowing for causality modelling by fMRI investigations which informs future fMRI results (but see caveat below).”

(Response) We thank the positive comments.

Comment 1. “One real potential strength of the MS is the fact that the authors utilize the high temporal resolution of EEG signals for causality analysis, which makes a very important contribution to the field. However, as we learn in the supplement, only one patient with medial stimulation is analyzed. This is a huge drawback and I do not see a reason not to validate the method / findings in more patients with 4/5/6 es-TT sites including lateral stimulation.”

(Response) Thank you for this comment. This is also a question another reviewer raised.

We conducted this analysis in only one patient for following reasons on the initial submission. Specifically, this analysis requires that intracranial electrode should cover the ROI for the key nodes at single subject level. This patient had electrode coverage within all the brain parcellation of our interest (anterior cingulate cortex, prefrontal cortex, sensory-motor cortex, parietal and lateral temporal lobe).

A second reason was that the patient had multiple data sets (experiments) available, giving us substantially more samples. Importantly, Conditional Granger causality (CGC) analysis does *not* allow trial rejection for *some* of the channels in the dataset per trials because the analysis uses data from all channels (in other words, it is *not* a pair-wise connectivity

measure) . For example, we cannot perform the analysis on any trials whose data was rejected in any of the channels. So for the CGC analysis including many channels, it requires fairly large number of trials.

We have further looked into the patients in whom multiple experiments were available with electrode coverage in the ROIs of interest. We found that PT384 (lateral and medial stimulation) and PT515 (only lateral stimulation) datasets were potentially suited to the analysis and ran the same Conditional Granger Causality analyses and presented. The additional results are now presented in Fig. 6 and Supplementary Fig. S7.

Of note, the electrode placement was different in each patient for clinical reason. So even in the same ROI group, the actual recording location differs across patients. The number of es-TT runs also varied. These factors (including difference in trial numbers as mentioned above) as well as individual difference could potentially influence the CGC results. However, the new results in fact confirmed the findings presented in the initial submission. ACC, OFC and PFC are heavily connected bidirectionally (but asymmetrically) during medial amygdala stimulation. ACC - temporal lobe connectivity was again detected as in the initial manuscript.

Compared to the medial amygdala group stimulation, Conditional Granger Causality applied to the lateral group stimulation showed marked differences in ACC – PFC (dorsolateral part) connectivity (greatly reduced). On the other hand, ACC – OFC connectivity remained as prominent as with medial group stimulation.

We revised texts in the corresponding part of methods (page 26 , line 697 - 699), results (page 12 line 269 - 270, line 276 - 279 and line 283 - 286) and discussion sections (page 14 line 329 - 330).

***Comment 2.** “The MS fails short in addressing lateralization issues. One has to expect, that an unilateral epileptic pathology does change the connectivity at a brain wide level and it will do so in a lateralized way. Therefore, I do not agree with the basic assumption of the MS, that lateralization does not play a role. Please provide evidence for that assumption!”*

(Response) This is an important point related to ones also raised by other reviewers. We have now conducted laterality analyses; please see the response to points 3 from Reviewer 1, and points 6 and 7 from Reviewer 2 above.

The concern here that epileptic pathology would be expected to alter connectivity is very well taken. We agree that pathology may change the connectivity in brain-wide level. In our patient cohort, 12 out of 51 sessions were from the same side of pathology.

However, the influence from epileptic pathology is likely to be complex and different in each patient, and we think that effect may even involve both hemispheres. This is because we frequently observe that seizures which originates clearly from unilateral pathology often rapidly spread to the contra-lateral side and eventually both hemispheres are involved electrophysiologically. Although it is impossible to completely exclude the possibility of the effect from epileptic pathology in our study, we have excluded (for both stimulation and recordings) the contacts that were determined to be in the seizure onset zone to minimize the potential confounds from the pathology. As can be seen in Table S1, the structural abnormality in our patients differ vastly as does the clinical picture (such as onset and duration etc...), and often the location of epileptogenic pathology cannot even be lateralized. Given this large variability and complexity of clinical aspects of epilepsy in individual level, we are treating here as random noise whose nature we cannot adequately investigate. The point is mentioned in the limitation section in the discussion (page 16 line 386 - 393).

Comment 3. "The MS would also benefit a lot if differences regarding "structural imaging normal" and epileptic evidence patients would have been in focus and at least partially addressed. You present at several instances data of single subjects and you can pool 5 "structural imaging normal" subjects comparing to impacted ones. Can any sub-pathology characteristics be carved out? At least address this in the discussion."

(Response) This is also an important question. We carried out additional analyses comparing evoked potential (EP) magnitudes between a structural imaging normal group (N = 3) and a structural lesion group (N = 10), results from which are now presented in supplementary Fig. S11. There were significant differences between the two groups for N/P15 in precentral gyrus and for P150 in middle frontal gyrus. Given limited number of data points in the structural normal group, further subdivision in hemisphere could not be done. There was a tendency for having a clear EP peak in the structural normal group.

While we have quantified the patient groups suggested by the reviewer's comment, we note that larger sample sizes would be required to draw any conclusions, for two reasons. First,

there is extreme variability in structural imaging findings, as can be seen in the table S1 and typical for our patient population. Second, there is a very small number of patients in the structural normal group (also consistent with our patient population). On the other hand, given large variability of imaging findings, individually specific contributions will be substantially averaged out – the results we present are thus ones that are robust to these individual differences. The issue on the sub-groupings is discussed in discussion section (page 16 line 386 - 393).

Supplementary Figure S11. ROI averaged intracranial EPs waveform grouped by structural MRI findings. (a) EPs magnitude distributions mapped on the MNI brain for N/P15 and P150 for structural normal and abnormal group. **(b)** Group averaged es-TT response mapped on to MNI brain. Red line and area: Overall averaged waveform and its se (Structural MRI normal). Black line and area: Overall averaged waveform and its se (Structural MRI abnormal). Red dots on the waveform indicate significantly different N/P15 (dots at 15 ms) or P150 component (dots at 150 ms) of EPs between two conditions ($P < 0.05$, Bonferroni correction). Only shows ROIs with number of contacts > 20 for each condition. Number of valid contacts within the ROI are also shown. Only ROIs that showed significant difference are shown. Medial and lateral group stimulation were combined for the analyses.

Comment 4. *“Are any interactions of duration or onset of epilepsy found?”*

(Response) This is a good question also raised by another reviewer. We performed additional analysis to see if there is any significant impact of duration of epilepsy on the amygdala’s connectivity contrasting (< 4 yr) vs (>20 yr) (as suggested by another reviewer).

Results are presented in supplementary figure S10. Statistical tests (ttests with Bonferroni correction) indicated that there were a few significant differences between the two groups (but much more restricted than the full set of results presented in the main text in the initial submission), specifically in middle temporal gyrus (p150). Given the small sample size within the subgroups and the vastly varying clinical presentations at individual level, much larger cohorts of subjects will be required to answer this question reliably.

We additionally examined whether the response patterns differed among various sub-grouping of patients (by the duration of epilepsy or structural abnormality in the MRI). The results are presented below (Fig. S12). This scatterplot matrix shows the es-TT responses are well correlated across various sub-groupings, suggesting relatively stable results irrespective of epilepsy duration or structural brain abnormality. The issue on the sub-groupings is discussed in discussion section (page 16 line 386 - 393).

Supplementary Figure S11. ROI averaged intracranial EPs waveform grouped by duration of epilepsy. (a) EPs magnitude distributions mapped on the MNI brain for N/P15 and P150 for epilepsy duration over 20 years and <4 years group. (b) Group averaged es-TT response mapped on to MNI brain. (b) ROI EPs that showed significant difference between groups. Red line and area: Overall averaged waveform and its se (Epilepsy duration < 4 yr). Black line and area: Overall averaged waveform and its se (Epilepsy duration > 20 yr). Red dots on the waveform indicate significantly different N/P15 (dots at 15 ms) or P150 component (dots at 150 ms) of EPs between two conditions ($P < 0.05$, Bonferroni correction). Only shows ROIs with number of contacts > 20 for each condition. Number of valid contacts within the ROI are also shown. Only ROI that showed significant difference is shown. Medial and lateral group stimulations were combined for the analyses.

Correlations of es-TT response among sub-groups. Each dot represents response magnitude (in sd unit) averaged within a ROI. Pearson correlation values were calculated and indicated by colored box.

Supplementary Figure S12. es-TT response magnitude distribution averaged within each structural ROIs and comparisons across different sub-groupings. Pearson correlation values (r) larger than 0.25 were indicated by colored axis-box.

Comment 5. “Even having in mind, that you report very valuable intracranial human data it remains unclear which patients and averages you present where in certain parts of the MS. Do we see a grand average of patients and controls? Do you mix right vs. left pathologies? Do you treat all indications identical.”

(Response) Thank you for your comment. We made the point clearer in the texts.

We presented EPs from individual es-TT run’s in Fig. 1a as an example. Otherwise, we present results of grand averages across subjects for each ROIs, with the sole exception of the Conditional Granger Causality results that was done per subject because of the reason mentioned in our response to comment 1. We treated patients equally because of the vast variability of individual clinical presentation in our patients. As the reviewer implies, there might be interesting contrasts between conditions, such as, right pathology – right stimulation, left pathology – right stimulation, left pathology – left stimulation and right pathology – left stimulation. Although these are interesting comparisons, since we do not

have controlled right and left pathologies in the same structures and given limited number of patient subjects we have, we did treat right and left pathologies equally.

Just like other intracranial electrophysiological research in humans, all our subjects were epilepsy patients, and no healthy control was available.

We, however, did perform a separate analysis focusing on the presence or absence of psychiatric condition (specifically, depression and anxiety) and present this in Supplemental Fig. S9. Results showed significant differences in dorsomedial and dorsolateral prefrontal cortices (DMPFC and DLPFC including superior and middle frontal gyrus), superior and middle temporal gyri and angular gyrus ($P < 0.05$, ttest, Bonferroni corrected). Importantly, prefrontal cortex showed significantly greater es-TT response in depression/anxiety group whereas middle and superior temporal gyri showed significantly weaker response in that group.

Altered amygdala-prefrontal (especially to ventromedial and dorsolateral part of PFC) connectivity in psychiatric condition has been reported, and our analysis add new information on the direction and timing of the amygdala connectivity. Though this result is extremely interesting, we remain cautious to draw rigid conclusion from this small sample data. This point is discussed in the paper (results section page 11 line 256 - 265 and discussion section page 16 line 382 – 385).

Figure S9. Amygdala stimulation es-TT response in patients with depression and/or anxiety. (a) ROIs that showed significant difference between patients with and without either depression or anxiety disorder ($N = 5$ and 8 , respectively) is color-coded and shown. ROIs in white mesh indicated non-significant ROIs. **(b)** es-TT response averaged over all ROIs for N/P15 (left) and P150 (right). Mean and SE are shown. Number of valid contacts are also shown. **(c)** Same as (b), but ROIs that showed significant difference ($P < 0.05$, Bonferroni correction). Medial and lateral group stimulations were combined for the analyses.

Comment 6. *“It is a quite heavy job to try to follow subject numbers throughout the MS. Please at least add to Table 1 a column indicating the epilepsy patients, sorted not by ID but by the groups (structural normal, epilepsy findings) and by examination type esTT, es-fMRI. Add type of treatment / medication.”*

(Response) Sorry, we understand the paper became complex. The information is now added to Supplementary Table S1. Table S1 is now sorted by the following three keys. 1st, Structural findings. 2nd, Duration of epilepsy and 3ed, Experiment type.

Comment 7. *“In the same regard it is unclear at several instances showing average results what was averaged. Please add consequently this information.”*

(Response) We have now made this point clear throughout the manuscript.

Comment 8. *“In addition, it would be of great help to the reader to follow the complex manuscript if the common thread were represented better overall.”*

(Response) We have checked this point throughout the manuscript.

Comment 9. *“Table 1: Some points already mentioned above. Moreover, please add medial / lateral stimulation sides per subject and provide at bottom average + sde across the parameters wherever possible.”*

(Response) This information has now been added in Table S1.

Comment 10. *“Which accuracy do you report? I expect that you used balanced accuracy?”*

(Response) The accuracy we reported for es-TT data was *classification accuracy*, and not balanced accuracy. We think that the classification results we reported are valid for two reasons. First, the degree of imbalance in the class was not severe. N medial = 2055 and N lateral = 1655. $2055/(2055+1655) = 0.554$. Second, we report accuracy of shuffled dataset while maintaining the ratio of the class label of original data. This point is added in method section (page 24 line 608 - 610). The term “prediction accuracy” used in the original Fig. 2f

and g was not a formal term and we changed it to “classification accuracy” in the revised manuscript.

Comment 11. *“P7,l177 20 runs and 10 does not match numbers given in Fig. 3”*

(Response) This is because some patient underwent stimulation of the same contacts combination multiple times; the different numbers refer to different stimulation sessions in the same patient (page 8 line 164 - 165).

Comment 12. *“P8,l194: I assume you mean Figure 3e”*

(Response) Fig. 3d is correct, but see our response below.

Comment 13. *“P8,l196: I assume you mean Figure 3f, g does not exist:”*

(Response) We are sorry for this mistake. Fig. 3f is correct as the reviewer indicated. This figure has been corrected. We also corrected the typo referring to Fig. 3e (page 9, line 181 and page 9 line 183).

Comment 14. *“P8,l205: In the text you used OF, in the figure OFC.”*

(Response) Thank you for catching this. This has been corrected. We used OFC to indicate orbitofrontal cortex (anatomically defined) and OF to indicate orbitofrontal group (ROI grouping we used, page 9, line 190 - 191) even these indicates the same anatomical structures. Since OF is a part of prefrontal cortex (ventromedial and ventrolateral part), we now use IPFC for indicating lateral prefrontal cortex group (PFC was used in the initial submission) to distinguish them more clearly (page 9 line 191 – 192). Now OF is used in the figures for es-TT results.

Comment 15. *“P9,l211/212: OF and OFC used for the same”*

(Response) This is related to the previous comment. This has been corrected (page 9 line 197 and 198). In the context of analysis on ROI groups, we use OF, otherwise we use OFC.

Comment 15. *“P9,I222: Why was no UMAP performed for the early N/P15 component? Please provide rationale or add it. Looks even more clear across the groups than for the P150 component.”*

(Response) Thank you for your suggestion. We have additionally performed UMAP analysis on the N/P15 component and presented in the Fig. 4d. One caveat for applying this analysis on the N/P15 was that evoked potentials in the early time window showed inconsistent polarity. This was not a problem for EPs amplitude and latency analysis since these metrics were not affected by the polarity. For the new UMAP analysis requested by the reviewer for the N/P15 component, we first detected the polarity of the N/P15 for each es-TT run and aligned the polarity by flipping the EPs waveform so that the early component always had positive polarity, and ran the same UMAP analysis as done for P150. Result from this analysis is added in results section (page 10 line 207 - 212).

Comment 16. *“P11,I276: please provide reference for the Leiden community algorithm.”*

(Response) This information is in the method section in the original manuscript (reference # 53, in page 28, line 709). We added this information in the main text too (page 11, line 296).

Comment 17. *“P17,I437: Please add version of Freesurfer.”*

(Response) We used FreeSurfer version 7.2.0. This information is added in Method section (page 19, line 471).

Comment 18. *“P23,I592: Coregistration to subject’S T1 and MNI? These statement is unclear.”*

(Response) We apologize; the sentence was incomplete and now corrected as follows. “ ... **Coregistration of subject’s T1 images to the MNI template and coregistration of subject’s BOLD images to the MNI template were performed.**” (page 26, line 650 - 652).

Comment 19. *“P23,I593: Do you make any use of tCompCor, CSF and white matter? From the text it appears that you calculate it but do not use it.”*

(Response) The reviewer is correct. Although these signals were calculated, they were not used (except for framewise displacement). We have deleted this sentence accordingly to avoid any confusion (page 26 line 652 - 654).

Comment 20. *“Do the patients differ in their medication / treatment regime? Add that info to table 1 and comment please.”*

(Response) This information has now been added in the new columns in table S1.

Comment 21. *“As one can see nicely in Fig. S3 the rank of latencies across the brain structure groups are highly comparable for NP15 and P150. Please discuss mechanistically how that can come along for latencies in 10th ms but also preserved in 1xx ms.”*

(Response) This is a great point and relates to the mechanisms of es-TT response generation. N/P15 (early component) most likely reflects excitatory response of pyramidal cells at sites that has structural connection to the stimulated sites. Later P150 likely reflects suppression of the local brain tissue triggered by the initial excitation (as reflected by N/P15) (Creutzfeldt et al., 1966; Keller et al., 2014; Kobayashi et al., 2017). P150 thus may have more complex mechanism than the early component and might include more functional connectivity aspect (rather than structural connection). However, as discussed above, N/P15 and P150 components are *not* independent events. And it is likely that the latency relationship is relatively well preserved. This mechanistic discussion is now included in the manuscript (page 15 line 354 - 359).

Comment 22. *‘Of note, the methods part is not very detailed. It is too superficial at several points like hardware descriptions, system software’s of the devices, algorithm parametrization, software versions etc. Particular using R please provide the hardware architecture and software versions of R and the libraries used.’*

(Response) We added these information (system software version, algorithm parametrization and R version) in the Methods section (page 19 line 454, page 24 line 597 - 601, 605 - 608).

Comment 23. *“Were any structural non-linearities found between pre-implantation and post-implantation? Does that effect registration / (assumed) electrode position.”*

(Response) We have also been aware of this issue, namely, co-registration between pre- and post-implantation structural MRIs. Particularly, in the presence of intracranial electrodes on the pial surface in the post-implantation images, the brain shift is a non-linear phenomenon that needs meticulous caution. We applied control-points based non-linear image morphing using thin-plate splines (TPS) (Oya et al., 2009). The post-implantation MRI volume was non-linearly transformed so that it matches the pre-implantation MRI volume. We used an advanced version of the above TPS morphing implementation in 3-D space. The electrode locations in the post-implantation images were non-linearly transferred onto pre-implantation images using this technique. This is explained in Methods section in the original manuscript (page 19 line 467 - 469).

Comment 24. *“Cross-validation is mentioned in the main text, not in the methods.”*

(Response) The information on the cross-validation is now added in the Methods section (page 24 line 608 - 610).

Minor points:

Comment 25. *“iEEG is not introduced before usage.”*

(Response) It is now introduced in the results section (page 6, line 105).

Comment 26. *“Please carefully check the usage of EEG vs iEEG (this is what you did) vs ECoG as well as partial (regressed against what) vs pearsons correlation. The terms are not to be used synomy.”*

(Response) Thank you for pointing this out. ECoG is used for the signal from the pial surface grids and strips, and SEEG is used for the signal from depth electrodes. iEEG is signal from intracranial electrode (combined ECoG and SEEG). We made sure the correct terminology was used in the manuscript.

Regarding the es-fMRI correlation measure, we used partial correlation to obtain brain's connectivity matrix. The partial correlations between the ROI pair used for this analysis were calculated while controlling the effect of *all the rest of variables (ROIs)*. Pearson correlation was then used to find the *similarity between partial correlation matrices* in different conditions (es-ON and es-Off).

Comment 27. *“Unify abbreviations e.g. esTT or es-TT.”*

(Response) This has been corrected. Now, the abbreviation is “es-TT”. Also we have added an abbreviation table.

Comment 28. *“P3,152: I suggest to delete EXPERIMENTAL animals.”*

(Response) Change has been made in the texts as suggested.

Comment 29. *“P5,1103: Please add “in cortical regions recorded” because iEEG is dominated by cortical regions.”*

(Response) We agree and change has been made in the texts as suggested (page 5 line 93).

Comment 30. *“P21,1536: incomplete sentence”*

(Response) We are sorry for this mistake. We have corrected this sentence (page 23 line 577 - 578).

Comment 31. *“Typographical and punctuation issues throughout the MS.”*

(Response) We made sure that these are corrected throughout the manuscript.

Points on Figures:

Comment 32. *“Fig. 1:B: PT 418 brain surface photography not mentioned in the legend.”*

(Response) We added the information on this picture in the legend (legend for Fig. 1b)

Comment 33. *“Fig. 2 b,c I suggest to shift insets b and c (classifications) to the end following the descriptions in the text. Also introduce specific titles for b and c.”*

(Response) Thank for the suggestions. We agree with these points. Fig. 2 panel b and c are moved to the end in the figure as panel f and g.

Comment 34. *“Fig. 2e/g ROIS are hardly visible. Please enhance e.g. lighter grey for the brain. Moreover, one thought: in red e encodes lateral > medial but red also encodes in g medial > lateral. One could use two different colors here (see e.g. usage in Fig. 3).”*

(Response) We agree with the reviewer. We have revised the figures according to the reviewer’s suggestion.

Comment 35. *“Fig. 3a: What does (Rt: 7, Lt: 9) encode? Not mentioned in the legends.”*

(Response) Number of stimulation points in each side of amygdala . We now mentioned this in the figure legend for Fig 3a.

Comment 36. *“Fig. 3b: legend: How can a single subject single voxel time course have a. error measure? I assume we see averages over given ROI? Please clarify.”*

(Response) The error bars shown in the Fig. 3b came from multiple electrical stimulation blocks, and not over given ROI. The standard errors of single voxel time series were

calculated over stimulation block (N = 10) in each es-fMRI run. This is now explained in the corresponding figure legend.

Comment 37. *“If the stimulation lasts till 30s how to explain signal reduction before stimulus ends?”*

(Response) We noticed that the stimulation period was not clear in the original Fig. 3b. Revised Fig. 3b now has stimulation period more clearly indicated with bars above x-axis. Associated legend has been revised. As shown in the figure 3 (b), the main response (either in positive or negative sign) quickly goes back to baseline after 30 sec.

There was a tendency for the reduction of the response after 15 – 20 seconds from the onset of es-ON period. This may reflect an adaptation mechanism observed in terms of connectivity, but this interesting topic is out of the scope of current manuscript and we would like to focus on the overall response here.

Comment 38. *“Fig. 4c: Here N = number of recording channels, in Fig. 5d N =number of contacts.”*

(Response) Thank you for pointing this out. The N should mean number of recording channels in the ROIs. We corrected this in the legend for Fig. 5d .

Comment 39. *“Fig. 6: Which stimulation and what are the patient characteristics shown here.”*

(Response) The figure 6 shows results from medial amygdala stimulation on PT511. This information was in the legend but not in the actual figure. We added this in the figure too. We now have additional Conditional Granger Causality results from different patients in the figure (Fig. 6 and Fig. S7).

References

- Alvarez, E.O., and Banzan, A.M. (2011). Functional lateralization of the baso-lateral amygdala neural circuits modulating the motivated exploratory behaviour in rats: Role of histamine. *Behavioural Brain Research* 218, 158–164.
- Angold, A., Costello, E.J., and Erkanli, A. (1999). Comorbidity. *Journal of Child Psychology and Psychiatry* 40, 57–87.
- Baas, D., Aleman, A., and Kahn, R.S. (2004). Lateralization of amygdala activation: a systematic review of functional neuroimaging studies. *Brain Research Reviews* 45, 96–103.
- Bickart, K.C., Hollenbeck, M.C., Barrett, L.F., and Dickerson, B.C. (2012). Intrinsic Amygdala–Cortical Functional Connectivity Predicts Social Network Size in Humans. *J. Neurosci.* 32, 14729–14741.
- Bzdok, D., Laird, A.R., Zilles, K., Fox, P.T., and Eickhoff, S.B. (2013). An investigation of the structural, connectional, and functional subspecialization in the human amygdala. *Human Brain Mapping* 34, 3247–3266.
- Cahill, L. (2006). Why sex matters for neuroscience. *Nat Rev Neurosci* 7, 477–484.
- Coleman-Mesches, K., and McGaugh, J.L. (1995). Differential involvement of the right and left amygdalae in expression of memory for aversively motivated training. *Brain Research* 670, 75–81.
- Creutzfeldt, O.D., Watanabe, S., and Lux, H.D. (1966). Relations between EEG phenomena and potentials of single cortical cells. I. Evoked responses after thalamic and epicortical stimulation. *Electroencephalography and Clinical Neurophysiology* 20, 1–18.
- Demeter, S., Rosene, D.L., and Van Hoesen, G.W. (1990). Fields of origin and pathways of the interhemispheric commissures in the temporal lobe of macaques. *J. Comp. Neurol.* 302, 29–53.
- Freeman, J.B., Stolier, R.M., Ingbreetsen, Z.A., and Hehman, E.A. (2014). Amygdala Responsivity to High-Level Social Information from Unseen Faces. *Journal of Neuroscience* 34, 10573–10581.
- Grayson, D.S., Bliss-Moreau, E., Machado, C.J., Bennett, J., Shen, K., Grant, K.A., Fair, D.A., and Amaral, D.G. (2016). The Rhesus Monkey Connectome Predicts Disrupted Functional Networks Resulting from Pharmacogenetic Inactivation of the Amygdala. *Neuron* 91, 453–466.
- Gupta, R., Koscik, T.R., Bechara, A., and Tranel, D. (2011). The amygdala and decision-making. *Neuropsychologia* 49, 760–766.

- Hardee, J.E., Thompson, J.C., and Puce, A. (2008). The left amygdala knows fear: laterality in the amygdala response to fearful eyes. *Social Cognitive and Affective Neuroscience* 3, 47–54.
- Keller, C.J., Honey, C.J., Mégevand, P., Entz, L., Ulbert, I., and Mehta, A.D. (2014). Mapping human brain networks with cortico-cortical evoked potentials. *Phil. Trans. R. Soc. B* 369, 20130528.
- Kilpatrick, L.A., Zald, D.H., Pardo, J.V., and Cahill, L.F. (2006). Sex-related differences in amygdala functional connectivity during resting conditions. *NeuroImage* 30, 452–461.
- Kobayashi, K., Matsumoto, R., Matsushashi, M., Usami, K., Shimotake, A., Kunieda, T., Kikuchi, T., Yoshida, K., Mikuni, N., Miyamoto, S., et al. (2017). High frequency activity overriding cortico-cortical evoked potentials reflects altered excitability in the human epileptic focus. *Clinical Neurophysiology* 128, 1673–1681.
- Kravitz, H.M., Schott, L.L., Joffe, H., Cyranowski, J.M., and Bromberger, J.T. (2014). Do anxiety symptoms predict major depressive disorder in midlife women? The Study of Women's Health Across the Nation (SWAN) Mental Health Study (MHS). *Psychol. Med.* 44, 2593–2602.
- Mishra, A., Rogers, B.P., Chen, L.M., and Gore, J.C. (2014). Functional connectivity-based parcellation of amygdala using self-organized mapping: A data driven approach. *Human Brain Mapping* 35, 1247–1260.
- Oya, H., Kawasaki, H., Dahdaleh, N.S., Wemmie, J.A., and Howard III, M.A. (2009). Stereotactic Atlas-Based Depth Electrode Localization in the Human Amygdala. *Stereotact Funct Neurosurg* 87, 219–228.
- Sergerie, K., Chochol, C., and Armony, J.L. (2008). The role of the amygdala in emotional processing: A quantitative meta-analysis of functional neuroimaging studies. *Neurosci. Biobehav. Rev.* 32, 811–830.
- Sullivan, R.M., Duchesne, A., Hussain, D., Waldron, J., and Laplante, F. (2009). Effects of unilateral amygdala dopamine depletion on behaviour in the elevated plus maze: Role of sex, hemisphere and retesting. *Behavioural Brain Research* 205, 115–122.
- Sylvester, C.M., Yu, Q., Srivastava, A.B., Marek, S., Zheng, A., Alexopoulos, D., Smyser, C.D., Shimony, J.S., Ortega, M., Dierker, D.L., et al. (2020). Individual-specific functional connectivity of the amygdala: A substrate for precision psychiatry. *Proc. Natl. Acad. Sci. U.S.A.* 117, 3808–3818.
- Tranel, D., and Bechara, A. (2009). Sex-related functional asymmetry of the amygdala: preliminary evidence using a case-matched lesion approach. *Neurocase* 15, 217–234.
- Wang, S., Yu, R., Tyszka, J.M., Zhen, S., Kovach, C., Sun, S., Huang, Y., Hurlmann, R., Ross, I.B., Chung, J.M., et al. (2017). The human amygdala parametrically encodes the intensity of specific facial emotions and their categorical ambiguity. *Nat Commun* 8, 14821.

Wilson, C.L., Isokawa, M., Babb, T.L., Crandall, P.H., Levesque, M.F., and Engel, J. (1991). Functional connections in the human temporal lobe. *Experimental Brain Research* 85, 174–187.

REVIEWERS' COMMENTS

Reviewer #1 (Remarks to the Author):

The authors have done a nice job conducting new analyses and revising the text in response to reviewer comments.

Reviewer #2 (Remarks to the Author):

In the revised manuscript from Sawada et al., the authors have performed additional analyses and edited the text to address the concerns raised by the initial round of reviews. The revised manuscript is greatly improved and remains an important achievement that I believe is ready for publication. While I appreciate the additional analyses included in the supplementary information, I would like to echo the authors' own statements that due to the sample size these analyses are likely underpowered. The intention of my comments about sex differences laterality, etc, was not to mine the data set beyond its capability, but to investigate unaccounted for sources of variation. The authors should be sure to indicate which analyses were planned in advance (and should be reflected in the sample size) and which were performed post hoc so the reader does not mis-interpret the study design.

Reviewer #3 (Remarks to the Author):

In my opinion the authors did an excellent job in addressing the issues raised by the referees during the first submission.

The MS has improved a lot, particular by adding more detailed data and analyses regarding additional aspects like laterality, gender etc.

As such I see it as a very important contribution to the amygdala and EEG community.

But personally I like the most the combination with fMR, i.e. combining electropysiological recording with fMRI establishing brain wide investigations via real causality analysis!

Some minor points still remain:

p2l30: IPFC not introduced

p4l61: „Structural connectivity studies show similar (but not identical) patterns ...“ having the rest of that paragraph in mind I would only say comparable, not similar.

p5l91: subsequent anatomical analysis ????

p6l117 number 2059 and 1667 does not match numbers in Fig. 1e!

p8l162-163: „The es-fMRI was conducted at rest (notask) with a block design with charge-balanced square pulse electrical stimulation at 9–12 mA current as described previously“ . [SEP]at rest (no task) ... one could call the es a task ... for which you used a block design. please clarify!

p8l166: Multiple comparison correction?

p15l347-348: „... lateral group stimulation“ ... does not make sense, Please rewrite.

p23l572-573: So finally how many EPs were used for which ROI, finally?

p23l577-578: latency definition for the peaks .. move this part up, before peak usage.

Some minor syntactical and typographical edits still remain.

Please write in-vivo consistently.

Point-by-point response to the reviewers.

The reviewers' comments are in italic font with quotation marks. Responses are in bold.

NCOMMS-21-36322A

("Mapping Effective Connectivity of Human Amygdala Subdivisions with Intracranial Stimulation")

Reviewer #1

"The authors have done a nice job conducting new analyses and revising the text in response to reviewer comments."

(Response) We thank the reviewer for the positive comments.

Reviewer #2

"In the revised manuscript from Sawada et al., the authors have performed additional analyses and edited the text to address the concerns raised by the initial round of reviews. The revised manuscript is greatly improved and remains an important achievement that I believe is ready for publication. While I appreciate the additional analyses included in the supplementary information, I would like to echo the authors' own statements that due to the sample size these analyses are likely underpowered. The intention of my comments about sex differences laterality, etc, was not to mine the data set beyond its capability, but to investigate unaccounted for sources of variation. The authors should be sure to indicate which analyses were planned in advance (and should be reflected in the sample size) and which were performed post hoc so the reader does not mis-interpret the study design."

(Response) Thank you for the thoughtful comments. We note in our revised paper which analyses were planned in advance to ensure clarity on this issue.

Reviewer #3

"In my opinion the authors did an excellent job in addressing the issues raised by the referees during the first submission. The MS has improved a lot, particular by adding more detailed data and analyses regarding additional aspects like laterality, gender etc.

As such I see it as a very important contribution to the amygdala and EEG community.

But personally I like the most the combination with fMR, i.e. combining electropysiological recording with fMRI establishing brain wide investigations via real causality analysis!"

(Response) Thank you for the detailed and constructive comments on our manuscript !

[Minor points]

Comment 1. *p2, line30: "IPFC not introduced"*

(Response) IPFC is now introduced in the main manuscript (page 7, line 206).

Comment 2. *p4, line61: "Structural connectivity studies show similar (but not identical) patterns ..." having the rest of that paragraph in mind I would only say comparable, not similar."*

(Response) We agree with the comment and revised the texts as suggested (page 3, line 67).

Comment 3. *p5, line91: "subsequent anatomical analysis ????"*

(Response) Thank you for indicating this point. Since the subsequent analyses we performed were indeed not "anatomical", we have deleted "anatomical" (page 4, line 101), and this part now reads as "subsequent analyses".

Comment 4. *p6, line117: "number 2059 and 1667 does not match numbers in Fig. 1e!"*

(Response) We apologize for this error. The numbers in the Fig.1e had not been updated correctly. We have now corrected these numbers in the Fig.1e and in the text (page 5, lines 129-130).

Comment 5. *p8, line 166: "Multiple comparison correction?"*

(Response) Multiple comparison corrections for es-fMRI results were done with cluster-based thresholding because voxel-wise thresholding (Bonferroni) is generally regarded as overly conservative for functional neuroimaging. The thresholding used was based on simulation using the latest version of AFNI's "3dClustSim" program that estimates false-positive probability of clusters based on spatial autocorrelation of the actual es-fMRI data. This point is explained in the manuscript in the method section (page 22, line 694 - 699). For the first-level analysis presented in Fig. 3b, we also used cluster-based threshold for the same reason (Figure 3b legend).

Comment 6. *p15, line 347-348: "... lateral group stimulation" ... does not make sense, Please rewrite."*

(Response) Thank you for indicating this point.

We have corrected this sentence as follows; "With lateral group stimulation, the N/P15 component showed a clear graded pattern of latency reflecting the timing of neural responses: the

pattern was such that responses were observed first in the anterior cingulate and orbitofrontal cortex (OF and CC group), followed by in the lateral prefrontal and lateral temporal cortex (IPFC and TL group), and the parietal cortex (PL group) showed slowest response within the ROI groups” (page 12, line 373-377).

Comment 7. p8, line162-163: “The es-fMRI was conducted at rest (notask) with a block design with charge-balanced square pulse electrical stimulation at 9–12 mA current as described previously“.at rest (no task) ... one could call the es a task ... for which you used a block design. please clarify!”

(Response) Because es-fMRI could also be done when subjects were engaged in a task, this sentence was to clarify that subjects were not engaged in any active task during the scanning. One could call electrical stimulation as a task as the reviewer indicated. Therefore, we deleted “at rest” in the manuscript and revised this sentence. The sentence now read as, “The es-fMRI was conducted without having subjects engaged in any active task using a block design (es-ON and es-OFF) with charge-balanced ...”, (page 6, line 176-177).

Comment 8. p23, line572-573: “So finally how many EPs were used for which ROI, finally?”

(Response) This information is presented in Supplementary figure S1, which shows number of EPs used within each ROI.

Comment 9. p23, line577-578: “latency definition for the peaks .. move this part up, before peak usage.”

(Response) We agree with this. This has been modified as suggested. The sentence that defines the latency is now moved up in page 18, line 583-585.

Comment 10. “Some minor syntactical and typographical edits still remain.

Please write in-vivo consistently.”

(Response) Thank you for indicating this. We have made sure the syntactical and typographical issues were corrected.